# High-throughput multiplexed serology via the mass-spectrometric analysis of isotopically barcoded beads

Alexandros P. Drainas[1,2] ✉, David R. McIlwain[3,4,5], Alec Dallas[1,2], Theresa Chu[3,4], Antonio Delgado-González [3,4], Maya Baron[1,2], Maria Angulo-Ibáñez[6], Angelica Trejo[4], Yunhao Bai[3,7], John W. Hickey[3,4,8], Guolan Lu[3,9], Scott Lu[10], Jesus Pineda-Ramirez[10], Khamal Anglin[10], Eugene T. Richardson [11,12], John C. Prostko[13], Edwin Frias[13], Venice Servellita [14], Noah Brazer[14], Charles Y. Chiu [14,15], Michael J. Peluso[16], Jeffrey N. Martin[10], Oliver F. Wirz [3], Tho D. Pham[3,17], Scott D. Boyd[3,18], J. Daniel Kelly[10,15,19,20,21], Julien Sage[1,2,23], Garry P. Nolan [3,23] ✉ & Xavier Rovira-Clavé [3,4,22,23] ✉

In serology, each sample is typically tested individually, one antigen at a time. This is costly and time consuming. Serology techniques should ideally allow recurrent measurements in parallel in small sample volumes and be inexpensive and fast. Here we show that mass cytometry can be used to scale up multiplexed serology testing by leveraging polystyrene beads uniformly loaded with combinations of stable isotopes. We generated 18,480 unique isotopically barcoded beads to simultaneously detect, in a single tube with 924 serum samples, the levels of immunoglobulins G and M against 19 proteins from SARS-CoV-2 (a total of 36,960 tests in 400 nl of sample volume and 30 µl of reaction volume). As a rapid, high-throughput and cost-effective technique, serology by mass cytometry may contribute to the effective management of public health emergencies originating from infectious diseases.

Serology immunoassays are powerful tools that measure specific antibodies in biological samples. These assays have a wide range of applicability, including monitoring vaccine efficacy[1–4], epidemiological surveillance[5], mapping viral transmission dynamics[6,7], infectious and autoimmune disease diagnosis[8,9], and drug discovery[10]. In situations with large number of samples requiring assessment across multiple targets[11], the scalability of current serology assays is a major bottleneck. There is a pressing need, highlighted by the coronavirus disease 2019 (COVID-19) pandemic, for precise, accurate, highly parallel, low-cost, fast and recurrent serology measurements that use small sample volumes.

Classic immunoassays such as the enzyme-linked immunosorbent assay (ELISA) are widely used because of simplicity, sensitivity and specificity. However, in ELISA, each sample is tested individually for one molecular target at a time, which severely hampers high-throughput applications due to cost and time constraints. In recent years, automation strategies and multiplexed serology techniques have increased throughput[12–25]. For example, fluorophore barcoded beads used in Luminex assays allow higher throughput by simultaneously detecting multiple analytes in a single sample[14,26]. However, the spectral overlap of fluorophores constrains the available number of barcodes and, thus, limits the scalability of fluorescent bead-based serology assays. An elegant strategy has recently ameliorated a major issue arising from spectral overlap in beads, nonlinear behaviour of multicolour Förster resonance energy transfer and resulting cascades, but the barcoding space of this method is currently limited to the generation of 580 distinguishable fluorophore combinations[27]. The maturity and widespread acceptance of bead-based serology assays have allowed these assays

to become industry standards. However, scalable paradigms capable of greatly surpassing existing barcoding limitations, to the range of tens of thousands of uniquely barcoded beads, would be a substantial breakthrough for research and clinical applications.

Mass cytometry is a mass-spectrometric, multiparameter and single-cell technology extensively used in the past decade for cell profiling[28]. Over 50 parameters are routinely analysed on each cell using antibodies conjugated to isotopically enriched rare earth metals[28]. The instrument provides mass spectra on the fly, one event at a time, at a speed of about 1,000 events per second. In mass cytometry, channel overlap is minimal, and thus, the theoretical barcoding space when 38 stable isotope channels are used is in the range of billions. These unique capabilities of a mass cytometer could be harnessed for high-throughput serology testing by assessing antibody binding to isotopically barcoded beads loaded with specific targets[29–31]. Bead-based assays for cytokine profiling via mass cytometry have been recently reported but are limited to the simultaneous detection of nine analytes[32]. This mismatch between the theoretical capability of mass cytometry for serology testing and its practical implementation is largely explained by the laborious experimental requirements for the production of isotope-loaded beads compatible with mass cytometry using current techniques[32]. Therefore, the vast potential of mass cytometry for serology testing has remained largely untapped.

Here, we report a practical, low cost, highly scalable bead-based multiplex serology testing using mass cytometry. We first show that polystyrene beads can be uniformly and robustly loaded postsynthesis with stable isotopes and can be reliably quantified via mass cytometry. We then illustrate the scalability of the method via the generation of 18,480 distinct isotopically barcoded beads. By harnessing this large barcoding space, we report the detection of antibody levels against multiple severe acute respiratory syndrome coronavirus 2 (SARS-CoV-2) proteins in multiple clinical serum and plasma samples simultaneously in a one-tube assay. This assay takes one operator approximately 8 h to complete for 924 samples, and no automation equipment is necessary. Our study demonstrates that mass cytometry with barcoded beads facilitates serology testing of multiple antigens in samples from large sample cohorts. This technology has the potential to contribute to management of public health and infectious disease pandemic response.

## Results

### A postsynthesis strategy for robust incorporation of stable isotopes to polystyrene beads

Two requirements are necessary to enable high-throughput bead-based assays via mass cytometry: (1) uniform incorporation of a high load of isotopes per bead and (2) a scalable strategy generating a large number of isotope-barcoded beads. Dispersion polymerization of isotope-containing polystyrene beads is a standard approach for synthesis of beads containing a high load of isotopes[33], but this procedure is not ideal for high-throughput barcode generation because each synthesis is performed individually (Supplementary Fig. 1a). Passive absorption of isotopes postsynthesis of the beads is an appealing strategy for barcoding due to being highly scalable, but in practice, isotope loading is non-uniform across beads, thus impairing data decoding (Supplementary Fig. 1a,b).

We reasoned that binding of isotope-conjugated biotinylated proteins to streptavidin-coated polystyrene beads would provide a high load of isotopes per bead and a uniform labelling, the latter understood as single beads in the population having similar isotope intensities (Fig. 1a and Supplementary Fig. 1a), while being compatible with microfluidic technologies for high-throughput combinatorial dispensing. We conjugated the stable isotope dysprosium 162 ($^{162}$Dy) to biotinylated bovine serum albumin (BSA) and loaded it to streptavidin-coated beads. Mass cytometry of these beads revealed a high load of isotopes per bead and a uniform labelling (Fig. 1b).

Next, we extended the approach to 37 additional stable isotopes, ranging in atomic mass from lanthanum 139 to ytterbium 176 ($^{176}$Yb) by generating beads with two types of combination of the isotopes. 'Odd' beads were loaded with the stable lanthanide isotopes having an odd mass number, and 'even' beads were loaded with the stable lanthanide isotopes having an even mass number (Fig. 1c). Using mass cytometry, we observed the expected positive signal for each isotope for both 'odd' and 'even' beads (Fig. 1d and Supplementary Fig. 1c), and the intensities for each isotope were similar despite the distinct transmission factors for each isotope typically observed in inductively coupled plasma mass spectrometry[34] (Fig. 1e and Supplementary Fig. 1d). The analysis of a pool consisting of beads loaded with distinct amounts of biotinylated protein carriers conjugated to $^{172}$Yb and $^{176}$Yb revealed the expected nine populations of beads (Supplementary Fig. 2), highlighting the tunability, flexibility and robustness of the system. Together, these analyses showed a uniform, tuneable and high load incorporation of 38 stable lanthanide isotopes to polystyrene beads postsynthesis and confirmed that this strategy is compatible with the loading of up to 19 distinct isotopes per bead.

### A high-throughput strategy to create thousands of isotope-barcoded polystyrene beads

There are two main components involved in high-throughput serology assay applications: (1) the ability to analyse multiple molecular targets per sample and (2) the ability to analyse large number of samples. To address both, we designed a double-barcoding strategy in which two sets of isotopes are present in each bead: one to identify the molecular target and another for identifying the sample (Extended Data Fig. 1a). Each set of isotopes provides a barcode, and thus, each bead has two barcodes (Extended Data Fig. 1b). Each of the two barcodes consists of $k$ isotopes from a total of $n$ isotopes, where $k = n/2$ for efficient generation of minimally redundant combinations and robust singlet detection[35]. The use of two barcodes, instead of a single barcode that encodes both sample and molecular target identification, is also convenient because it enables harnessing postsynthesis bead incorporation of isotopes in sequential loading steps (Extended Data Fig. 1c).

We created 18,480 barcodes using the double-barcode strategy using 18 isotopes, ranging from terbium 159 ($^{159}$Tb) to $^{176}$Yb. Each bead was first loaded with three out of the six isotopes from the first set ($^{159}$Tb to $^{164}$Dy) to generate 20 barcodes, pooled and then loaded with 6 out of the 12 isotopes from the second set (holmium 165 ($^{165}$Ho) to $^{176}$Yb) to generate 924 barcodes (20 × 924 = 18,480 barcodes) (Fig. 2a and Extended Data Fig. 1b). As approximately half of the beads are expected to be positive any given isotope, when analysed, each of the 18 isotopes showed the expected bimodal distribution in the global bead population (Supplementary Fig. 3). We developed an automatic debarcoding pipeline capable of identifying bead events with a correct double-barcode signature (Supplementary Fig. 4). From the analysis of three independent batches of barcoded beads, with a mean of $1.88 \pm 0.31 \times 10^6$ events per run, we accurately debarcoded 72.9% ± 10.97% of bead events with a correct signature of 6 out of 12 isotopes from the second set, and from those, 85.29% ± 8.91% of bead events had a correct signature of 3 out of 6 isotopes from the first set (Supplementary Fig. 5). Accurately debarcoded bead events grouped in one of the 20 groups from the first barcode set (Fig. 2b and Supplementary Figs. 6 and 7), and within each group, beads further separated in 924 groups from the second barcode set (Fig. 2c and Supplementary Figs. 8 and 9). Bead counts for each of the 924 barcodes from the second barcode set followed a normal distribution (Supplementary Fig. 10), with 100% recovery of the 18,480 barcodes (Supplementary Fig. 11). Together, these data demonstrated high-throughput generation of thousands of isotope-barcoded beads and their robust identification from a pooled mixture using mass cytometry.

Identification of thousands of barcoded beads from a mixture offers a versatile system for the design of different types of multiplexed

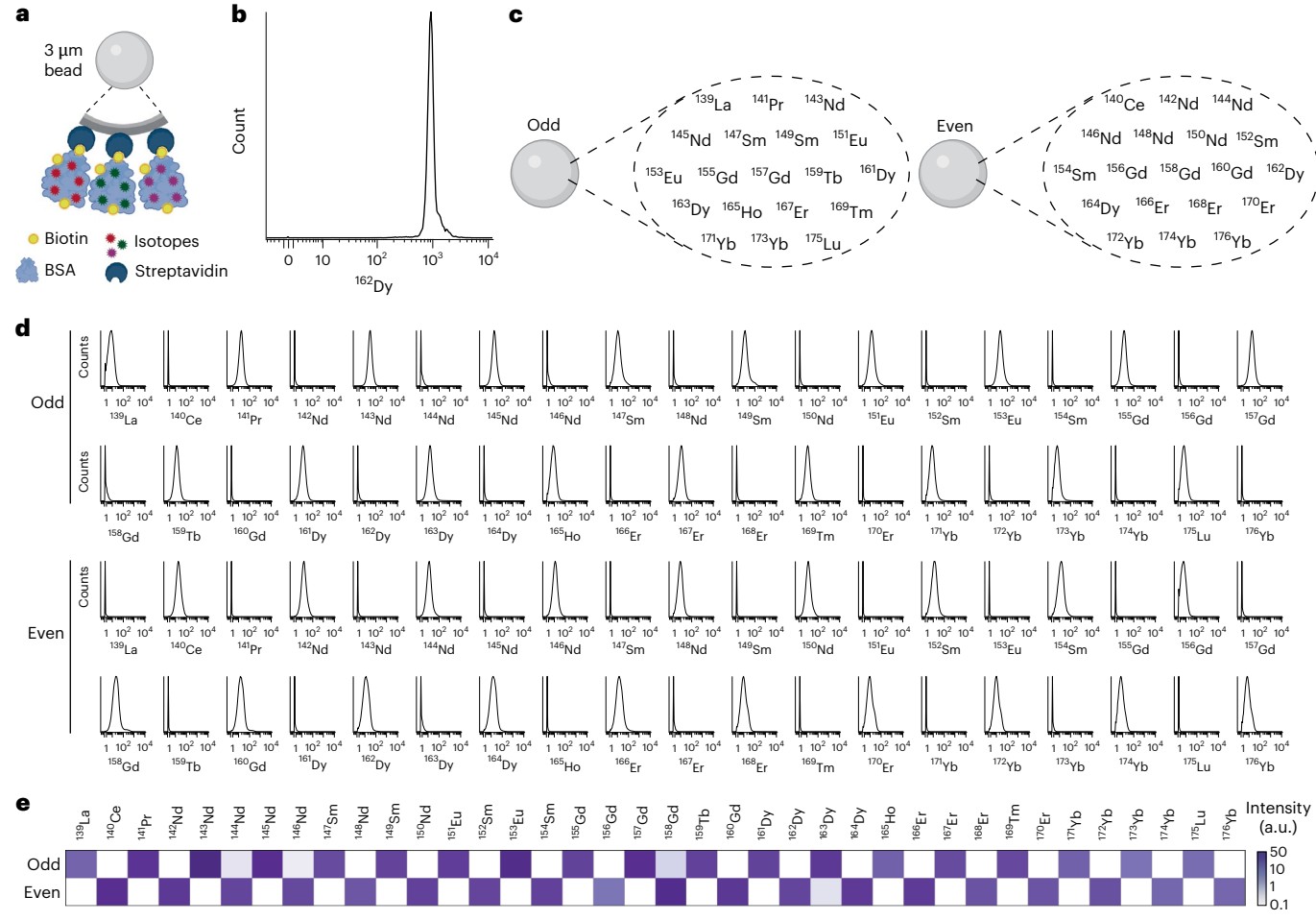

**Fig. 1 | Postsynthesis strategy for incorporation of stable isotopes on polystyrene beads. a**, A schematic representation of streptavidin-coated polystyrene beads loaded with isotope-conjugated biotinylated protein carrier. **b**, A representative histogram of $^{162}$Dy intensity per bead in mono-labelled beads. The $^{162}$Dy per bead was measured by mass cytometry. **c**, A schematic representation of beads loaded with 'odd' and 'even' isotopes. **d**, The representative histograms of isotope intensity per bead for 'odd' and 'even' beads. **e**, A heat map of isotope intensities for 'odd' and 'even'. The colour shows the median of the normalized isotope intensities for odd and even bead populations.

serology assay with unique capabilities. As a demonstration of these capabilities, we designed two systems: the first for detection of serum antibodies using a very low sample volumes (Fig. 3) and the second for rapid, wash-free, high-throughput detection of serum antibodies (Fig. 4).

### An assay for detection of plasma antibodies using low sample volume

We designed a two-bead selection system to detect the presence of specific antibodies in low volumes of human plasma (Fig. 3a). In this system, one bead is barcoded with a predefined combination of stable isotopes and loaded with a biotinylated antigen of interest (for example, a purified protein or a protein subunit). The other bead is magnetic and is conjugated to antihuman immunoglobulins. Cross-linking of barcoded beads to magnetic beads occurs only in the presence of antibodies recognizing target antigens (Extended Data Fig. 2). For example, in an assay leveraging the two-bead selection system, each clinical plasma sample is incubated in individual wells with barcoded beads loaded with distinct antigens to allow host antibodies to bind to antigens on beads. The barcoded beads are then washed to remove any unbound antibody and pooled across all samples to create a baseline bead mixture. The magnetic beads are then added providing an opportunity for cross-linking of magnetic beads to barcoded beads via antihuman immunoglobulins when host antibodies are present.

Washes are performed using a magnet to retain only those barcoded beads from host antibody positive samples, and the flowthrough is then analysed by mass cytometry. For each barcode, the ratio of the bead count in the flowthrough and the baseline bead mixture provides an estimate of host antibody levels, with low ratio values implying the presence of host antibodies specific to the antigen in a given sample.

As a demonstration of this method, we first incubated barcoded beads loaded with the recombinant SARS-CoV-2 S1 subunit of the spike protein (spike S1) with 20 plasma samples from COVID-19 convalescent donors and 20 negative control plasma samples collected before the emergence of COVID-19, all diluted 300-fold. The 40 distinct types of barcoded bead were similarly represented in the pooled baseline mixture, which is collected before magnetic bead selection (Fig. 3b,c). Consistent with expected patterns of serum immunity, after selection, all barcoded beads incubated with pre-COVID-19 pandemic samples were enriched (Fig. 3b,c), and most barcoded beads incubated with patient samples positive for SARS-CoV-2 were depleted (Fig. 3c). Two of the SARS-CoV-2-positive samples resembled pre-COVID-19 samples (Fig. 3c). Consistently, independent ELISA analysis of the SARS-CoV-2-positive samples showed high anti-spike S1 levels for all samples except the two behaving similarly to pre-COVID-19 samples (Fig. 3d), suggesting these two patient samples may have low levels of SARS-CoV-2 antibodies despite being from individuals with prior SARS-CoV-2-positive polymerase chain reaction (PCR) tests.

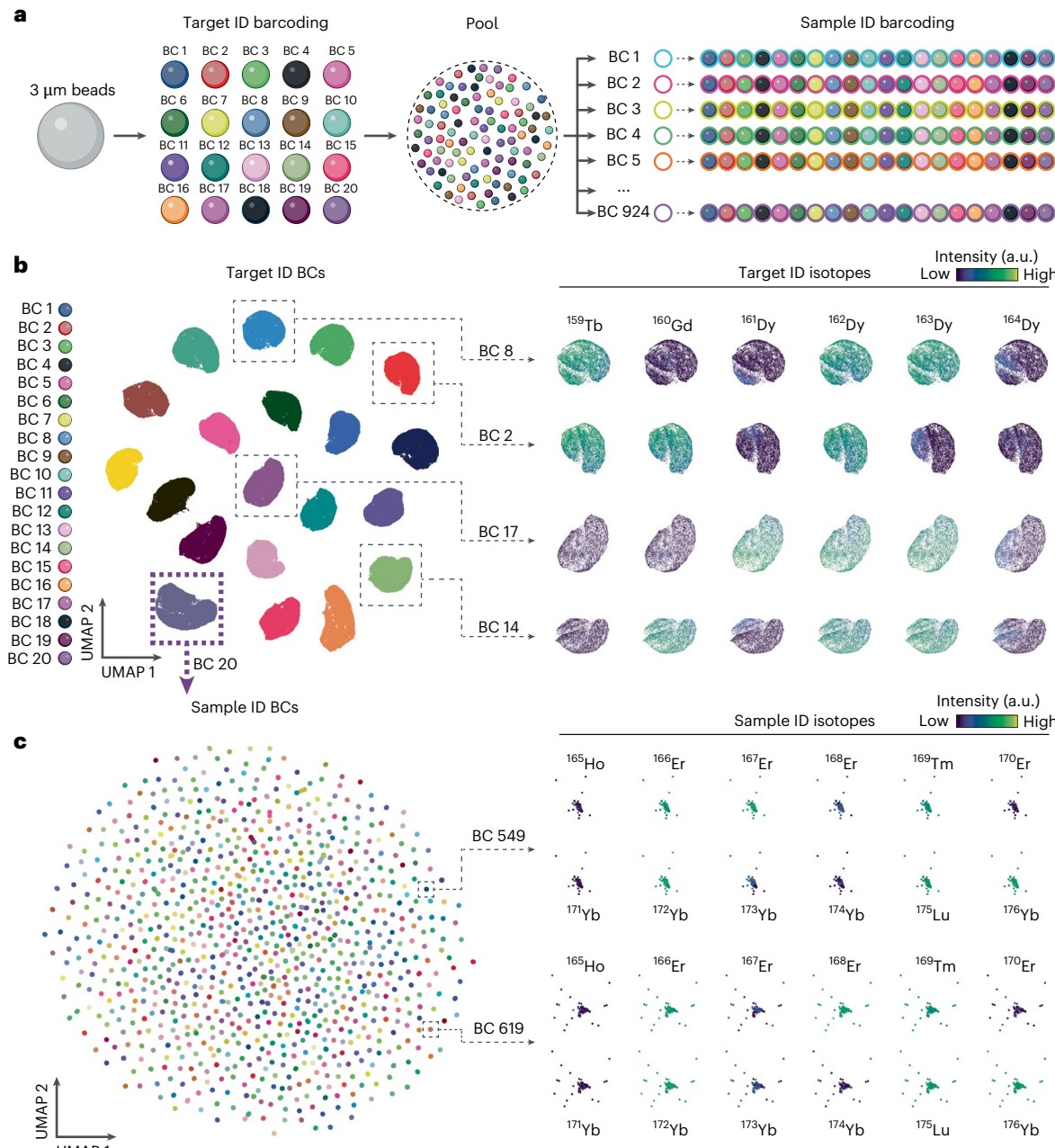

**Fig. 2 | Double-barcoding strategy enabled generation of 18,480 isotope-barcoded beads. a**, A schematic representation of the workflow for the generation of isotope-barcoded beads. For target ID barcoding, the beads are labelled with 3 of the 6 target ID isotopes ($^{159}$Tb to $^{164}$Dy) to generate 20 barcodes (BCs). For sample ID barcoding, the beads are labelled with 6 of the 12 sample ID isotopes ($^{165}$Ho to $^{176}$Yb) to generate 924 BCs. **b**, Left: a uniform manifold approximation and projection (UMAP) of beads grouped by the intensity of the target ID isotopes and coloured by target ID BC. Right: representative groups coloured by the intensity of the target ID isotopes. Each dot represents a bead. The data are from three independent experiments. **c**, Left: a UMAP of beads from BC 20 label in **b** grouped by the intensity of the sample ID isotopes and coloured by sample ID BC. Right: representative BCs 549 and 619 coloured by the intensity of the sample ID isotopes. Each dot represents a bead. The data are from three independent experiments.

We observed similar results in three independent experiments (Supplementary Fig. 12). Furthermore, we observed similar results when samples were analysed individually in an independent flow cytometry assay rather than pooled together in the mass cytometry assay (Supplementary Fig. 13a). We additionally observed proper alignment of a multiplexed assessment of the levels of antibodies against SARS-CoV-2 spike S1, the receptor binding domain of spike S1 (spike S1 RBD), and the nucleocapsid protein to the results of an ELISA assay (Supplementary Fig. 13b). To gauge sensitivity of the assay, we incubated beads loaded with spike S1 with increasing concentrations of a monoclonal antibody

against spike S1 and observed detection down to 100 pg ml$^{-1}$ for this particular antibody-antigen interaction (Supplementary Fig. 14). These results show that the mass cytometry results align with the results obtained with ELISA and flow cytometry, validating the two-bead system for antibody detection in patient plasma samples.

We next leveraged the extensive barcoding space of the isotope-tagged beads to analyse an additional set of 39 plasma samples from individuals who had tested positive for SARS-CoV-2 by PCR and 55 plasma samples collected before the COVID-19 pandemic. Each sample was diluted 1:300, 1:3,000 and 1:30,000, which is equivalent to using

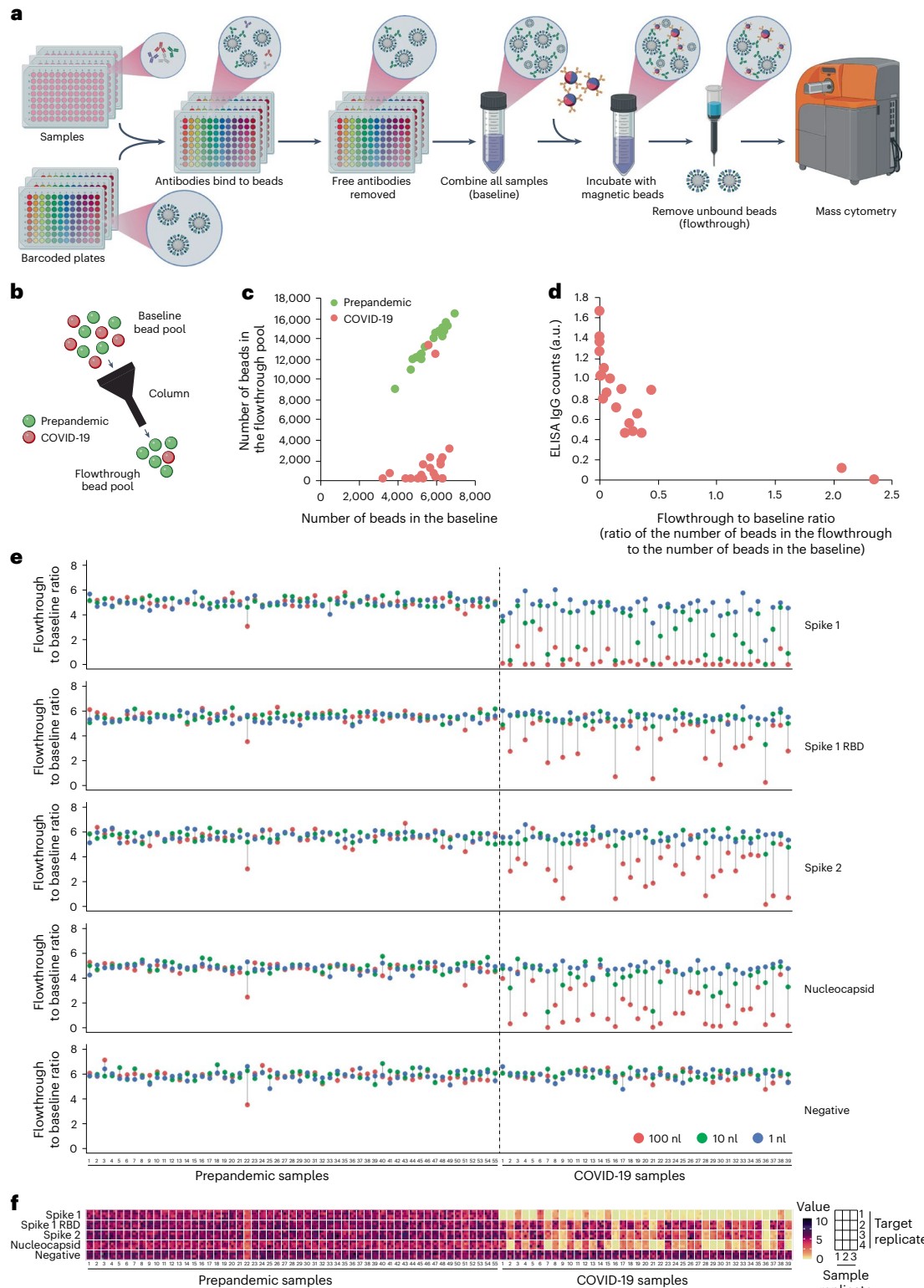

**Fig. 3 | Two-bead strategy to detect specific antibodies in low volumes of plasma. a**, A schematic representation of the two-bead strategy for detection of antibodies in low volumes of plasma. **b**, A schematic representation of the baseline and flowthrough. The beads bound by anti-IgG magnetic beads are retained in the column, and the unbound beads are collected in the flow through. **c**, A dot plot of the number of barcoded beads in the baseline versus number in the flowthrough for 20 serum samples collected before the COVID-19 pandemic (green) and 20 samples from patients who had tested positive for COVID-19 (red). **d**, A comparison of ELISA (*y* axis) versus the two-bead assay (*x* axis) in the 20 COVID-19-positive samples shown in **c**. The two-bead assay result is shown as the ratio of the number of beads in the flowthrough to the number of beads in the baseline. Note that the two samples in **c** that behave as prepandemic samples also show low ELISA values. **e**, A two-bead assay on a set of 39 COVID-19-positive samples and 55 prepandemic samples. The dot colour indicates the sample volume: 100 nl is red, 10 nl is green and 1 nl is blue. Each dot is the mean of 12 replicates (3 sample volume replicates and 4 target replicates per sample volume). The targets were: COVID-19 spike S1, spike S1 RBD, spike S2, nucleocapsid and negative (no target). **f**, A heat map of the values of the 12 replicates for each red dot shown in **e**.

100, 10 and 1 nl of sample, respectively, in a 30 µl reaction volume. We incubated the samples with barcoded beads loaded with recombinant SARS-CoV-2 spike S1, spike S1 RBD, the S2 subunit of the spike protein (spike S2) and the nucleocapsid protein. Each sample and dilution were quantified 12 times using three sample replicates and four target replicates by calculating the ratio of the bead count in the flowthrough and the baseline bead mixture in each replicate (Extended Data Fig. 3). The two-bead selection system was able to distinguish all 39 patients with COVID-19 samples from negative controls with 100% success rate using 100 nl of sample volume (Supplementary Fig. 15). In some samples, antibodies against spike S1 were detected using only 1 nl of sample (Fig. 3e; samples 21 and 36, spike S1, blue dots). The assay also revealed patient-specific variability in SARS-CoV-2 antibody profiles. For instance, some samples had high levels of antibodies against spike S1 but not against spike S1 RBD, spike S2 or the nucleocapsid (Fig. 3e; samples 14 and 23, red dots). Others had high levels of antibodies against spike S1, spike S1 RBD, spike S2 and nucleocapsid (Fig. 3e; samples 16 and 28, red dots). Lastly some had high levels of antibodies against spike S1 and nucleocapsid but not spike S1 RBD and spike S2 (Fig. 3e; samples 13 and 15, red dots). For each sample, each of the 12 replicates had similar values (Fig. 3f and Supplementary Fig. 16), and the results correlated well in two independently performed assays (Supplementary Fig. 17a). In addition, the levels of antibodies against SARS-CoV-2 spike S1 and nucleocapsid had a strong correlation with independent ELISA assays conducted on the same set of samples (Supplementary Fig. 17b). These data show that a two-bead strategy can detect host antibodies in low volume samples, down to 1 nl, and can provide unique profiles of the patient's antibody responses against multiple antigens.

## A high-throughput, wash-free assay for multiplexed detection of antibodies

Secondary antibodies are advantageous for serology assays due to ease of use and have been previously used in low-throughput bead-based mass cytometry assays[32]. We validated the compatibility of secondary antibodies conjugated to isotopes for the quantification of host immunoglobulins bound to the isotope-barcoded beads using positive and negative controls. Spike S1-loaded, isotope-barcoded beads were incubated with dilutions of a plasma sample from an individual positive for SARS-CoV-2 antibodies and with plasma samples from eight individuals collected before the COVID-19 pandemic as negative controls. The beads were washed, pooled and incubated with antihuman IgG conjugated to gold nanoparticles. We observed a positive control signal well above the background (Supplementary Fig. 18, sample 16 versus samples 1–12) that spanned linearly from 8 µl to 62.5 nl in a 1:2 serial dilution series (Supplementary Fig. 18, samples 13–20). We observed minimal non-specific binding of the secondary anti-IgG antibody to beads (Supplementary Fig. 18, samples 1–4) and low non-specific binding of prepandemic samples (Supplementary Fig. 18, samples 5–12). These results validated that antibodies bound to antigen-loaded, isotope-barcoded beads can be readily detected via secondary antibodies.

A major factor limiting the scalability of serology assays such as ELISA and fluorescently barcoded bead-based assays is the need to perform multiple washes, one plate at a time, to remove unbound antibodies. Automation strategies can ameliorate the issue[36] but are not straightforward to implement. We designed a system consisting of consecutive fixation and denaturation steps to eliminate washing constraints for isotope-barcoded beads serology assays (Fig. 4a). In this wash-free system, antigen-loaded isotope-barcoded beads are first incubated with samples of interest to allow host antibodies to bind to bead loaded with antigens. Each well is then treated with paraformaldehyde (PFA) to fix host antibodies onto the beads. A subsequent sodium dodecyl sulphate (SDS) treatment denatures the bound and unbound immunoglobulins (Supplementary Fig. 19) without removing isotopes from the beads (Supplementary Fig. 20). The samples are all then pooled and washed. Pooled samples are then incubated with isotopically labelled secondary antibodies, and the analysis is completed by mass cytometry to reveal the level of bound host antibodies on each barcoded bead.

To demonstrate bead-based multiplex serology testing by mass cytometry at scale in a wash-free system, we simultaneously analysed the levels of IgG (Fig. 4b) and IgM (Fig. 4c) against 19 distinct SARS-CoV-2 protein variants (Supplementary Table 1) in 542 longitudinal serum samples collected from a cohort of SARS-CoV-2 infected households in San Francisco (Supplementary Table 2). As part of this study blood samples were collected longitudinally from index cases with positive SARS-CoV-2 tests and additional household members. Extensive epidemiological data were also captured that included prior SARS-CoV-2 vaccination status. Expectedly, anti-wild-type SARS-CoV-2 Trimer IgG levels increased over time across the cohort as a whole (Fig. 4d), and these increased levels were strongly related to both vaccination status and presence of a positive SARS-CoV-2 PCR test (Supplementary Fig. 21). In the same assay, we included triplicates or quadruplicates of 100 negative samples collected before the COVID-19 pandemic. The immunoglobulin levels against wild-type SARS-CoV-2 trimer were below a given threshold in 97.5% of the tests for IgG and in 100% of the tests for IgM (Supplementary Fig. 22). In the samples collected at day 28 from infected and vaccinated individuals of the San Francisco study, we observed that all 23 samples were above the threshold for IgG (Supplementary Fig. 23), and 17 of the 23 samples were above the threshold for IgM (Supplementary Fig. 24). The levels of IgG against the wild-type SARS-CoV-2 trimer obtained via mass cytometry correlated well with those obtained using Abbott AdviseDx SARS-CoV-2 IgG II (Architect) (Fig. 4e), an assay that received emergency use authorization by the Food and Drug Administration.

Because this assay provides internally controlled data on multiple targets for the same sample, it is feasible to make robust and direct comparisons within individuals across SARS-CoV-2 variants. We observed a high correlation on the antibody response against SARS-CoV-2 variants for most samples (Fig. 4f) but, interestingly, identified a subset of samples with unique responses to certain variants (Fig. 4f) that could reflect variation in antibody repertoire driven by patient or virus-related factors. For example, sample numbers 4,567, 4,582 and 4,599 showed lower IgG levels against the beta, gamma, delta and omicron variants compared with the wild-type and alpha variant, and sample number 4,642 showed lower IgG levels against the beta, gamma and omicron variants compared with the wild-type, alpha and delta variants (Fig. 4f and Supplementary Fig. 25).

---

**Fig. 4 | A high-throughput and wash-free strategy for multiplexed detection of antibodies. a**, A schematic representation of the wash-free strategy to detect antibodies against 20 targets in 924 samples in a single mass cytometry analysis. **b,c**, A heat map of IgG (**b**) and IgM (**c**) levels against 19 targeted proteins (rows) in 542 samples (columns) from the FIND study, which collected samples from individuals with confirmed SARS-CoV-2-positive tests along with additional household members. The samples are labelled by day of sample collection after positive symptoms for the household's index case (timepoint), PCR test and vaccination status. IgG and IgM levels were normalized to values of beads not loaded with a target antigen (negative). The samples and targets are hierarchically clustered by IgG or IgM levels. **d**, A boxplot of the IgG levels against the SARS-CoV-2 spike trimer per sample by timepoint. Each dot represents a sample. **e**, A dot plot of the levels of IgG against the SARS-CoV-2 spike S1 trimer obtained by mass cytometry versus those obtained using the Abbott AdviseDx SARS-CoV-2 IgG II assay. Each dot represents a sample. The orange line is a polynomial regression line of degree 2. $R^2 = 0.58$. **f**, A dot plot of the levels of IgG against the beta SARS-CoV-2 spike trimer versus the wild-type trimer quantified using mass cytometry. Each dot represents a sample. NTD, N-terminal domain.

These data show that bead-based multiplex serology testing by mass cytometry is quantitative, sensitive and specific. This assay is capable of generating data comparable to existing authorized tests but with massive improvements in scalability, performing 36,960 tests in one tube using 400 nl of sample volume in a 30 µl reaction volume.

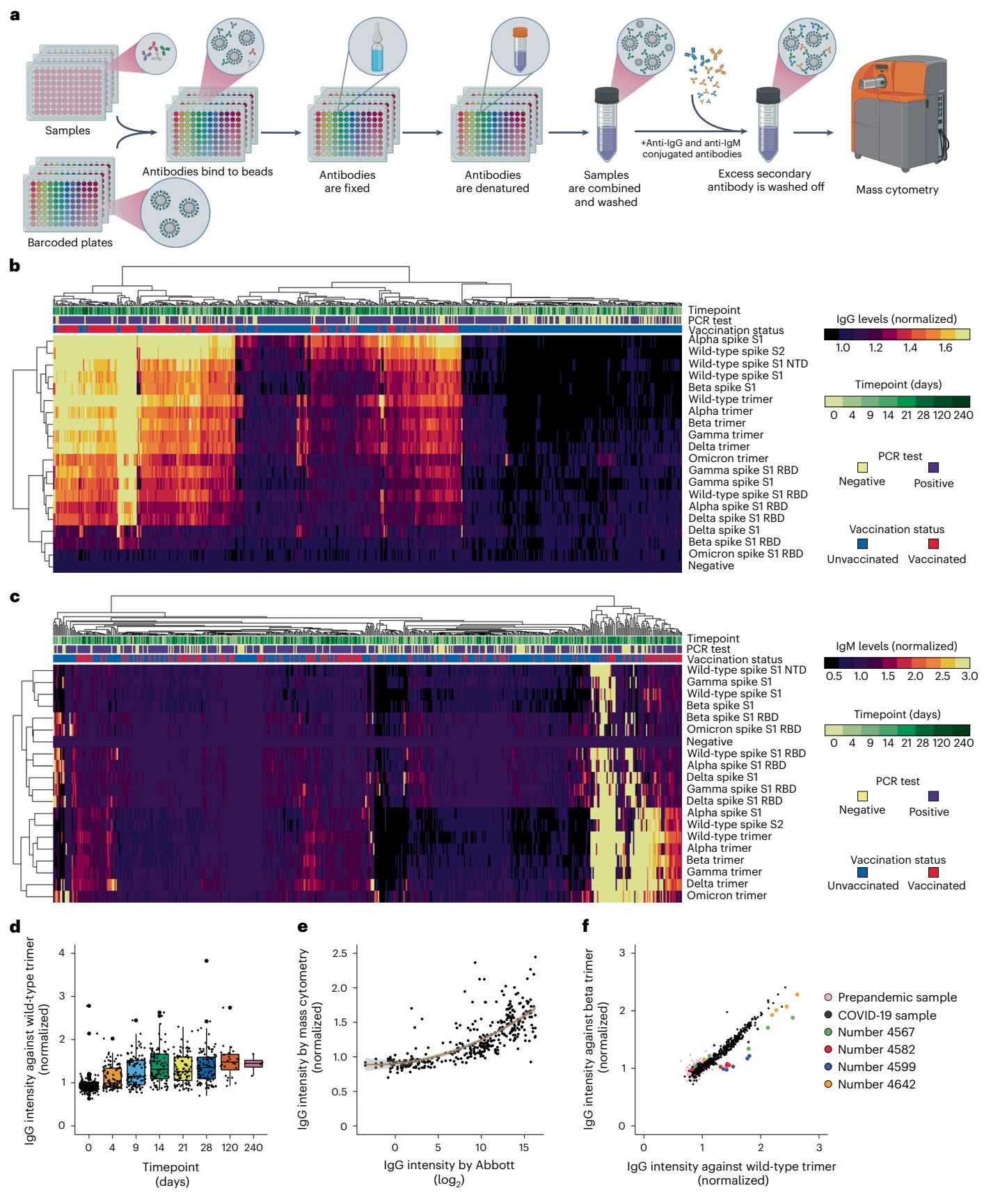

## Discussion

We report a highly scalable and multiplexed serology technology with readout via mass cytometry. We share the generation of 18,480 unique barcodes by incorporating distinct combinations of stable isotopes on polystyrene beads after synthesis. We also provide an automatic pipeline to debarcode the raw data without the need to input user-defined parameters. Using this technology and a bead-selection approach, we show the detection of IgGs against SARS-CoV-2 spike S1 using the equivalent of 100 nl of sample in a cohort of 55 prepandemic and 39 COVID-19 samples. In a subset of those samples, we show detection of IgGs using the equivalent of 10 and 1 nl of sample. Using secondary detection antibodies, we demonstrate the simultaneously analysis of IgG and IgM levels against 19 proteins in 924 samples in an assay that took one operator approximately 8 h to complete without the need of automation equipment. This is the equivalent of performing 36,960 ELISA-style serology assays in a single run. The ability to run tens of thousands of assays in parallel makes it a cost-effective technology (Supplementary Note 1). Moreover, compared with other strategies used in diagnostic labs such as ELISA and Luminex assays, the hands-on time of operators running the same number of assays is substantially reduced.

The assessment of humoral immunity in the FIND cohort allowed benchmarking of the serology assay via mass cytometry versus an immunoassay approved for clinical use. The data indicate that the mass cytometry assay can detect unique SARS-CoV-2 variant specific signatures in a robust fashion. This detection is made possible due to internal control resulting from pooling and simultaneous processing of all samples. This serology technology is immediately extensible into additional antigen variants from SARS-CoV-2 (ref. 37) and other targets by producing the required recombinant proteins. Comprehensive characterization of immunoglobulin subclasses coupled to longitudinal testing campaigns can reveal immune mechanisms that might inform vaccination campaigns, as illustrated by a preferential class switch to IgG4 after booster shots of an mRNA vaccine for SARS-CoV-2 (ref. 38). A future development could harness part of the remaining isotope space to include simultaneous assessment of IgG1, IgG2, IgG3, IgG4, IgM, IgA1 and IgA2 on each bead for deeper sample profiling. Other antibody locations with key biological relevance, such as the nasal mucosa[39] or gingival crevice[40], could be also interrogated by expanding the assay to additional sample sources.

From a technical standpoint, the current iteration of the isotope-based serology assay is mainly constrained by the number of available barcodes and acquisition speed. Over 50 isotopes are available off-the-shelf for mass cytometry[28], and we report the simultaneous use of 20 of them in the serology assay. The remaining isotopes could be harnessed to increase the number of barcodes, but careful validation should be performed given the distinct transmission factors for each isotope in inductively coupled plasma mass spectrometry. For instance, using 16 instead of 12 isotopes for sample ID would generate 12,870 combinations, and 8 instead of 6 isotopes for target ID would generate 70 combinations, for an aggregate of 900,900 barcodes. An increase in the number of barcodes should also be accompanied by further assay miniaturization and automation. The inductively coupled plasma time-of-flight mass spectrometry instrument used in this report has a relatively slow sample acquisition speed of 1,000 events per second, which limits the number of samples that can be analysed[41], but a higher speed could be achieved using alternative instruments[41], smaller particles and multiplex signal amplification systems[42,43]. The beads used in this study have a diameter variation of 3–3.4 μm, with larger beads having a higher number of streptavidin molecules on their surface. The use of beads with a narrower range of diameter variation will also improve the throughput of the assay by reducing the number of required beads for analysis. Moreover, the assembling of antigen panels is constrained by the scalability of protein purification. Our technology scales faster than the process of integrating new purified

proteins into the panel, thus posing a challenge in fully leveraging the multiplexing potential of the technology. The commercial availability of antigens ameliorates the issue, although extensive validation will be required when assembling new antigen panels.

In summary, we have designed and implemented a fast, high-throughput and cost-effective serology technology that opens new opportunities for profiling humoral immune responses at population scales. We envision that this technology will have a broad range of eventual applications, for instance in government-sponsored serology surveillance, drug-discovery efforts, monitoring of vaccination campaigns, diagnosis of autoimmune diseases and cancer and monitoring of immune correlates of protection during infectious disease outbreaks. These applications of mass cytometry-based serology will help reveal humoral immune mechanisms at a systems level, which may guide development of therapeutic strategies and enable prediction of clinical outcomes.

## Methods

### Clinical samples

Deidentified samples from the study[44] were used for Fig. 3c,d, comprising individuals with confirmed SARS-CoV-2 infection through PCR with reverse transcription (RT–PCR) testing. The samples used here were collected from COVID-19 convalescent plasma donors who donated plasma at the Stanford Blood Center between April 2020 and May 2020[44]. This study was approved by the Stanford University Institutional Review Board (protocols IRB-13952, IRB-48973 and IRB-55689).

Deidentified COVID-19-positive samples used for Fig. 3e,f were purchased from Discovery Life Sciences, comprising individuals with confirmed SARS-CoV-2 infection through RT–PCR testing.

Deidentified samples from FIND COVID were used for Fig. 4. FIND COVID is a Centers for Disease Control and Prevention (CDC)-funded longitudinal cohort of individuals recently diagnosed with SARS-CoV-2 infections in the San Francisco Bay Area, initiated in August 2020. The index cases were identified from individuals with a positive health provider-ordered SARS-CoV-2 nucleic acid amplification test at a UCSF-affiliated health facility. Index cases were enroled with uninfected household contacts and followed at five field visits over the first 4 weeks of illness. Study visits occurred on the day of enrolment and days 9, 14, 21 and 28 postindex symptom onsets. The blood samples were collected form index cases and household contacts at each study visit. Moreover, the participants self-collected anterior nasal swabs daily for the first 2 weeks following enrolment, then on days 17, 19, 21 and 28. All nasal specimens were used to quantify SARS-CoV-2 RNA through RT–PCR targeting nucleocapsid (N) and envelope (E) genes to determine SARS-CoV-2 infection status of household participants. The study was approved by the UCSF Institutional Review Board (protocol IRB number 20-30388) and given a designation of public health surveillance according to federal regulations as summarized in 45 CFR 46.102(d)(1)(2).

Deidentified prepandemic samples used for Figs. 3 and 4 were purchased from Solomon Park, comprising samples collected before November 2019 (the official start of the COVID-19 pandemic).

### Conjugation of BSA and antibodies to isotopes

Biotinylated BSA (Sigma, number A8549) was used as carrier protein and it was conjugated to isotope-chelated Maxpar X8 polymers (Fluidigm, number 201300). Briefly, 200 μg of biotinylated carrier proteins were reduced by tris(2-carboxyethyl)phosphine (TCEP) treatment (Thermo Fisher Scientific, number 77720) for 30 min at 37 °C. The reduced carrier proteins were reacted with isotope-chelated maleimide-containing polymers for 1.5 h at 37 °C and washed five times with 1× PBS. The protein concentration was quantified using a Nanodrop 2000 (Thermo Fisher Scientific, number ND-2000) and all conjugates were diluted to 0.5 g l$^{-1}$ in 1× PBS. Isotope conjugation was confirmed after every conjugation experiment by loading

isotope-conjugated BSA into beads and analysing them by mass cytometry. Antihuman IgM was conjugated to [115]In chelated Maxpar X8 polymers following the same procedure.

### Bead loading with isotope-conjugated carrier proteins

Streptavidin-coated polystyrene beads with a diameter of 3–3.4 µm (Spherotech, number SVP-30-5) were washed thrice in 1× PBS with 0.5% BSA and 0.001% Tween-20 (CSM-T). The beads where then incubated with 2.5 µg ml$^{-1}$ isotope-conjugated biotinylated carrier proteins for 30 min at room temperature, washed thrice in CSM-T and stored at 4 °C.

### Bead loading with isotopes by passive absorption

Streptavidin-coated polystyrene beads with a diameter of 3–3.4 µm (Spherotech, number SVP-30-5) were washed thrice in CSM-T. The beads where then incubated with 50 µM lanthanide chloride in 1× PBS for 1 h at room temperature, washed thrice in CSM-T and stored at 4 °C until analysis.

### Load proteins on beads for 924 assays

For 1 ml of streptavidin-coated polystyrene beads with a diameter of 3–3.4 µm (number SVP-30-5, Spherotech), 10 µl CSM with 0.1% Tween-20 (Sigma, number P1379) were added. The beads were washed thrice with 500 µl CSM-T by centrifuging at 2,800 relative centrifugal force (RCF) at room temperature. The supernatant was discarded. For each avi-tagged protein a 1:10 dilution was prepared (stock at 0.25 g l$^{-1}$) by adding water. The stock was quickly stored at −20 °C after usage. The beads were resuspended in 165 µl CSM-T and then split in 20 tubes (7.5 µl each) labelled from 1 to 20. To each tube, 7.5 µl of an avi-tagged protein dilution was added to the beads. The beads with the avi-tagged protein were incubated for 30 min at room temperature. The beads were washed thrice with 100 µl CSM-T by centrifuging at 2,800 RCF at room temperature. To each of the tubes 165 µl of CSM-T was added. The peads loaded with avi-tagged proteins were store at 4 °C until use.

### Generation of 20 barcodes for target ID

To create 20 barcodes, three isotope-conjugated biotinylated carrier proteins from a collection of six were distributed in wells of ten V-bottom 96-well plates (Costar, number 3363) containing 94 µl of CSM-T. The concentration of each individual three isotope-conjugated biotinylated carrier proteins was 1 µg ml$^{-1}$. After dispensing, 50 µl of the beads loaded with avi-tagged proteins were placed on each well. The plates were incubated for 30 min at room temperature. The beads were then washed thrice in CSM-T by centrifugation at 2,000 RCF at room temperature and stored at 4 °C until use.

### Generation of 924 barcodes for sample ID

High-throughput generation of barcodes was performed on a Tempest (Formulatrix). To create 924 barcodes, 6 isotope-conjugated biotinylated carrier proteins from a collection of 12 were distributed in wells of ten V-bottom 96-well plates (Costar, number 3363) containing 44 µl of CSM-T. After dispensing, the concentration of each individual six isotope-conjugated biotinylated carrier proteins was 1 µg ml$^{-1}$. The plates were manually transferred to a Bravo Automated liquid Handler Platform (Agilent) and 50 µl of the beads loaded with avi-tagged proteins were placed on each well. The plates were incubated for 30 min at room temperature. The beads were then washed thrice in CSM-T by centrifugation at 2,000 RCF at room temperature and stored at 4 °C until use.

### Two-bead assay by flow cytometry

For each sample, 5 µl of fluorescently labelled streptavidin polystyrene beads (Spherotech, numbers SVFP-0556-5 and SVFP-1068-5) were washed twice with CSM-T at a 1:1 ratio and pelleted by centrifuging at 21,000 RCF for 2 min after each wash and discarding the supernatant. Next, the beads were resuspended in 5 µl CSM-T per sample. Then, per

sample, 1 µl of biotinylated spike S1 at 0.25 µg µl$^{-1}$ (Sino Biological, number 40591-V27H-B) was added to 5 µl of beads and was incubated for 30 min at room temperature. The beads were washed three times by adding 100 µl of CSM-T, centrifuging at 21,000 RCF for 3 min and discarding the supernatant. Then the pellet was resuspended in 30 µl of CSM-T and transferred to a 96-well V-bottom plate (Costar, number 3363). Next, 10 µl of 1:10 diluted plasma in CSM-T (or plasma titrated from 1, to 1:1,000 µl) or an antibody titration was added to each condition. The beads were incubated for 30 min at room temperature and washed twice by adding 50 µl of CSM-T, centrifuging at 2,800 RCF for 3 min and discarding the supernatant. The beads were then resuspended in 100 µl of CSM-T. In each well of the plate, 2 µl of Magnetic-IgG beads (RayBiotech, number 801-101-1) were added. The beads were incubated for 30 min at room temperature with mixing by pipetting every 10 min. Subsequently, another 100 µl of CSM-T was added to each well and the plate was placed on a 96-well magnetic separator plate for 2 min to attach the magnetic beads. The unbound beads in solution were collected and added to another well. The collected beads were washed three times by adding 100 µl of CSM-T, centrifuging at 2,800 RCF for 3 min and discarding the supernatant. Finally, the beads were resuspended in 100 µl of CSM-T and analysed using a CytoFlex flow cytometer (Beckman Coulter). For each sample, acquisition was based on a specific volume, either 20 or 50 µl. Fluorescent Blue and Nile Red beads were initially identified and gated using forward and side scatter channels and subsequently gated on the basis of their fluorescence signal. This volume-based acquisition approach enabled the counting of the total number of beads in each sample volume. The counts obtained were then analysed and plotted using R.

### Two-bead assay by mass cytometry

The beads from the barcoded 96-well V-bottom plates (Costar, number 3363) loaded with biotinylated spike S1 (Sino Biological, number 40591-V27H-B) as described above were pelleted by centrifuging at 2,800 RCF, and the supernatant was discarded and then resuspended in 30 µl of CSM-T. Patient plasma was diluted in CSM-T 1:10, and 10 µl of each sample was mixed with the barcoded beads. The plate was incubated for 30 min at room temperature. The beads were washed twice by adding 100 µl of CSM-T, centrifuging at 2,800 RCF for 3 min and discarding the supernatant. Then, 100 µl of diluted spike S1 protein (2 ng µl$^{-1}$) in CSM-T (Sino Biological, number 40591-V02H) was added to each well and incubated for 10 min at room temperature. The plate was centrifuged at 2,800 RCF, the supernatant was discarded and another 50 µl of spike protein (0.2 ng µl$^{-1}$) in CSM-T was added. All samples were combined in a reservoir and added to a 50 ml tube and spun down at 2,800 RCF. Then, 10% of the beads were collected and labelled as 'baseline'. The samples were then diluted in 2.5× the number of samples in CSM-T (for 40 samples, we add 100 µl of CSM-T). We then added magnetic-activated cell sorting anti-IgG microbeads (Miltenyi Biotec, number 130-047-501) to a 1:2 ratio and incubated at room temperature for 30 min. The magnetic-activated cell sorting LS columns (Miltenyi Biotec, number 130-042-401) were prewashed with 1 ml CSM-T. The samples were diluted ten times and added to a LS column−1 ml of diluted sample was added to one LS column−and flowthrough was collected in a 5 ml protein-low bind tube (Fisher Scientific, number 0030122356). Another 4 ml was added and the flowthrough was collected. The beads from the flowthrough were pelleted at 2,800 RCF, and the supernatant was discarded. The beads were resuspended in 1 ml CSM-T and then then moved to a 1.5 ml protein-low bind tube (Fisher Scientific, number 022431081) and pelleted at 21,000 RCF. The beads were washed by adding 1 ml of CSM-T, centrifuging at 21,000 RCF for 3 min and discarding the supernatant. The beads were then washed three times with 0.5 ml H$_2$O before loading into CyTOF. The beads were analysed using a CyTOF 2 mass cytometer (Fluidigm) at a rate of 500 events per second. The focus of this analysis was on isotopes within the mass range of 159–176 atomic mass units.

## Wash-free assay by flow cytometry

The beads in 96-well V-bottom plates (Costar, number 3363) loaded with biotinylated spike S1 (Sino Biological, number 40591-V27H-B) were pelleted by centrifuging at 2,800 RCF, and the supernatant was discarded and the then resuspended in 30 μl of CSM-T. Spike S1 was detected by a human anti-spike S1 (Invivogen, number srbd-mab1) or patient plasma diluted 1:10 and 10 μl of each sample was mixed with the loaded beads. The plate was incubated for 30 min at room temperature. Next, 40 μl of PFA (3.2%) was added to each well of the respective samples and incubated for 15 min. Then, 80 μl of 4% SDS was added and incubated for 15 min. The plate was centrifuged at 2,800 RCF, and the supernatant from the beads was discarded. The wells were washed three times by adding 150 μl of CSM-T, centrifuging at 2,800 RCF for 3 min and discarding the supernatant. Next 5 μl of (1:200 dilution) of anti-IgG-647 (Thermo Fischer Scientific, number A21445) was added to each well. The beads were incubated at room temperature for 15 min. The wells were then washed three times by adding 150 μl of CSM-T, centrifuging at 2,800 RCF for 3 min and discarding the supernatant. Each well was resuspended in 150 μl of CSM-T and analysed by a Cyto-Flex flow cytometer (Beckman Coulter). For each sample, a volume of 50 μl was analysed. The beads were first distinguished and gated using forward and side scatter channels, followed by analysis based on the fluorescence intensity of anti-IgG conjugated with Alexa Fluor 647. Subsequent data analysis was performed utilizing R software.

## Wash-free assay by mass cytometry

The beads from the barcoded 96-well V-bottom plates (Costar, number 3363) loaded with biotinylated spike S1 (Sino Biological, number 40591-V27H-B) or 19 distinct SARS-CoV-2 protein variants (Supplementary Table 1) were pelleted by centrifuging at 2,800 RCF, and the supernatant was discarded and the then resuspended in 30 μl of CSM-T. Patient plasma were diluted 1:10, and 10 μl of each sample was mixed with the loaded beads. The plate was incubated for 30 min at room temperature. Next, 40 μl PFA (3.2%) was added to each well of the respective samples and incubated for 15 min. Then, 80 μl of 4% SDS was added and incubated for 15 min. Then, all the samples were combined in a reservoir by pipetting and then added to a 50 ml tube. The beads were washed three times with by adding 25 ml of CSM-T, centrifuging at 2,800 RCF for 3 min and discarding the supernatant. Next, the beads were stained with 25 ml CSM-T containing 256 ng l$^{-1}$ gold anti-IgG antibody (Nanoprobes, number 2053) and 256 ng l$^{-1}$ $^{115}$In-conjugated anti-IgM antibody (BioLegend, number 314502) for 30 min. The beads were pelleted at 2,800 RCF, resuspended in 200 μl CSM-T and transferred to a protein-low bind Eppendorf tube (Fisher Scientific, number 022431081). The samples were washed thrice with 200 μl of CSM-T and pelleted by centrifuging at 21,000 RCF after each wash and discarding the supernatant. The beads were then washed three times with 0.5 ml H$_2$O before loading into CyTOF. The beads were analysed using a CyTOF 2 mass cytometer (Fluidigm), at a rate of 500 events per second. The focus of this analysis was on isotopes within the mass range of 159–176, 115 and 197 atomic mass units. A step-by-step illustration of the assay and a document with the numbers to control and optimize the stoichiometry of biotin to streptavidin at the population level are described in Supplementary Notes 2 and 3.

## ELISA to detect anti-SARS-CoV-2 antibodies in plasma samples

The ELISA protocol performed for Fig. 3d and Supplementary Fig. 13b in this study was previously described[44,45]. In short, 96-well Corning Costar high binding plates were coated with SARS-CoV-2 spike S1, RBD or N protein in PBS at a concentration of 0.1 μg per well overnight and at 4 °C. The plates were then blocked using PBS–Tween-20 containing 3% non-fat milk powder for 1 h. The plasma samples from COVID-19 convalescent plasma donors were incubated at a dilution of 1:100 in dilution buffer (PBS-T containing 1% non-fat milk powder) for 1 h. Anti-SARS-CoV-2 IgG antibodies were detected using

horseradish peroxidase-conjugated goat antihuman IgG (Thermo Fisher, number 62-8420; 1:5,000 dilution). The plates were developed using 3,3′,5,5′-tetramethylbenzidine. The reaction was stopped by adding sulfuric acid after 12 min. The optical density was measured at 450 nm, and blank control values were subtracted. All the samples were tested twice in independent experiments. For Supplementary Fig. 17b, we used the ELISA kits: RAPID MAX SARS-CoV-2 Spike S1 Human IgG ELISA Kit (BioLegend, catalogue number 447809) and LEGEND MAX SARS-CoV-2 Nucleocapsid Human IgG ELISA Kit (BioLegend, catalogue number 448107). The protocol was implemented in strict accordance with the manufacturer's instructions with 500 nl of plasma sample.

## Abbott AdviseDx SARS-CoV-2 IgG II assay

SARS-CoV-2-specific antibodies were determined using the immunoassay AdviseDx SARS-CoV-2 IgG II for qualitative and semiquantitative detection of IgG antibodies to the Spike S1 RBD protein according to the manufacturer's specifications. Serologic analyses for this study using the Abbott AdviseDx SARS-CoV-2 IgG II (Architect) were performed under protocols approved by the UCSF Institutional Review Board (IRB number 11-05519).

## Automatic bead debarcoding

The beads were debarcoded using a custom R script. Briefly, all flow cytometry standard data files of a CyTOF run were concatenated. The events were then arcsine transformed with a cofactor of 5 and scaled using min-max normalization. The data for each isotope were plotted in a histogram to confirm a bimodal distribution. The signal intensities for each isotope were automatically thresholded using auto_thresh of the autothresholdr R package (version 1.3.2) with IJDefault method. A vector of thresholded intensities was used to transform the detected events to a binary matrix, in which '0' means signal lower than the threshold and '1' means signal higher than the threshold. Target and sample IDs were assigned following the target ID key (Supplementary Table 3) and sample ID key (Supplementary Table 4). The events with target and sample ID were selected for downstream analysis.

## Data visualization

The plots were created using the ggplot2 R package[46]. For the boxplots: the centre is the median, the minima and maxima bound of box are the 25th and 75th percentiles, respectively, the whiskers extend from the minima and maxima bounds of box to the largest value no further than 1.5 times the interquartile range, and the outliers are shown as dots. Figures 1a, 3a and 4a, Extended Data Figs. 1–3, Supplementary Figs. 1a, 2a, 3, 4a and 18a, and Supplementary Note 2 were created in part using BioRender.com. All figures were prepared using Illustrator (Adobe).

## Reporting summary

Further information on research design is available in the Nature Portfolio Reporting Summary linked to this article.

# Data availability

The data supporting the findings of the study are provided within the article and its Supplementary Information. Raw data related to this paper are available via Zenodo at https://doi.org/10.5281/zenodo.10822263 (ref. 47).

# Code availability

Code and a debarcoding demo related to this paper are available via Zenodo at https://doi.org/10.5281/zenodo.10822263 (ref. 47).

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

## Acknowledgements

We thank all the members of the Sage and Nolan labs for their help throughout this study. We thank A. Ray and N. Wang for their help with administration and R. Nissan for technical contributions. We thank S. Saydah, M. Briggs-Hagen and C. M. Midgley from the Coronavirus and Other Respiratory Viruses Division, National Center for Immunization and Respiratory Diseases, Centers for Disease Control and Prevention, USA. X.R.-C. discloses support for the research described in this study from the EMBO postdoctoral fellowship (ALTF 300-2017), the Spanish Ministry of Science Generación de Conocimiento Grant MOMIC (PID2023-152897OA-I00 funded by MICIU/AEI/10.13039/501100011033 and by FEDER/UE), the Ramon y Cajal Fellowship (RYC2022-037277-I funded by MICIU/AEI/10.13039/501100011033 and by FSE+) and the ERC Starting Grant SpaceClones (grant agreement number 101117905). A.P.D. discloses support for the research described in this study from the Tobacco-Related Disease Research Program (T30FT0824). M.A.-I. discloses support for the research described in this study from a SCI Fellowship Award. G.L. discloses support for the research described in this study from the NIH (K99CA267171). J.S. and G.P.N. disclose support for the research described in this study from the National Institutes of Health (3U19AI100627-09S2). J.D.K. discloses support for the research described in this study from a CDC Broad Agency Announcement (75D30120C08009). C.Y.C. discloses support for the research described in this study from a CDC project (75D30122C014367).

## Author contributions

A.P.D., J.S., G.P.N. and X.R.-C. conceived the project. A.P.D., D.R.M. and X.R.-C. designed the experiments. A.P.D., A.D., T.C., A.D.-G., M.B., M.A.-I., A.T., J.W.H., O.F.W., T.D.P. and X.R.-C. performed the experiments and acquired the data. A.P.D. and X.R.-C. analysed the data. Y.B., G.L. and X.R.-C. wrote the debarcoding and sample processing code. S.L., J.P.-R., K.A., E.T.R., V.S., N.B., C.Y.C., M.J.P., J.N.M. and J.D.K. provided the samples and data from the FIND study. J.C.P. and E.F. performed the Abbott testing work. A.P.D., D.R.M., S.D.B., J.D.K., J.S., G.P.N. and X.R.-C. supervised the project. A.P.D., J.S., G.P.N. and X.R.-C. acquired the funding. X.R.-C. wrote the original draft. A.P.D., D.R.M., J.S., G.P.N. and X.R.-C. wrote and edited the paper with input from all authors.

## Competing interests

A.P.D., J.S., G.P.N. and X.R.-C. have filed patent applications that cover some aspects of the technology described in this article. The patent with application number PCT/US2022/011620 was filed at the United States, Canada and European Patent Office. A.P.D., D.R.M., J.S., G.P.N. and X.R.-C. are cofounders and shareholders of Capture Bioscience, Inc. M.A.-I. is shareholder of Capture Bioscience, Inc. The other authors declare no competing interests.

## Additional information

**Extended data** is available for this paper at https://doi.org/10.1038/s41551-025-01349-0.

**Correspondence and requests for materials** should be addressed to Alexandros P. Drainas, Garry P. Nolan or Xavier Rovira-Clavé.

¹Department of Pediatrics, Stanford University, Stanford, CA, USA. ²Department of Genetics, Stanford University, Stanford, CA, USA. ³Department of Pathology, Stanford University, Stanford, CA, USA. ⁴Department of Microbiology and Immunology, Stanford University, Stanford, CA, USA. ⁵Department of Microbiology and Immunology, University of Nevada Reno, Reno, NV, USA. ⁶Department of Medicine, Stanford University, Stanford, CA, USA. ⁷Department of Chemistry, Stanford University, Stanford, CA, USA. ⁸Department of Biomedical Engineering, Duke University, Durham, NC, USA. ⁹Otolaryngology, Stanford University, Stanford, CA, USA. ¹⁰Department of Epidemiology and Biostatistics, University of California San Francisco, San Francisco, CA, USA. ¹¹Department of Global Health and Social Medicine, Harvard Medical School, Boston, MA, USA. ¹²Department of Medicine, Brigham and Women's Hospital, Boston, MA, USA. ¹³Applied Research and Technology, Abbott Laboratories Inc., Abbott Park, IL, USA. ¹⁴Department of Laboratory Medicine, Infectious Diseases and Global Medicine, University of California San Francisco, San Francisco, CA, USA. ¹⁵Department of Medicine, Infectious Diseases and Global Medicine, University of California San Francisco, San Francisco, CA, USA. ¹⁶Division of HIV, Infectious Diseases and Global Medicine, University of California San Francisco, San Francisco, CA, USA. ¹⁷Stanford Blood Center, Palo Alto, CA, USA. ¹⁸Sean N. Parker Center for Allergy and Asthma Research, Stanford University, Stanford, CA, USA. ¹⁹Institute for Global Health Sciences, University of California San Francisco, San Francisco, CA, USA. ²⁰F.I. Proctor Foundation, University of California San Francisco, San Francisco, CA, USA. ²¹San Francisco Veterans Affairs Medical Center, San Francisco, CA, USA. ²²Institute for Bioengineering of Catalonia, Barcelona Institute of Science and Technology, Barcelona, Spain. ²³These authors jointly supervised this work: Julien Sage, Garry P. Nolan, Xavier Rovira-Clavé. ✉e-mail: drainas@stanford.edu; gnolan@stanford.edu; xrovirac@ibecbarcelona.eu

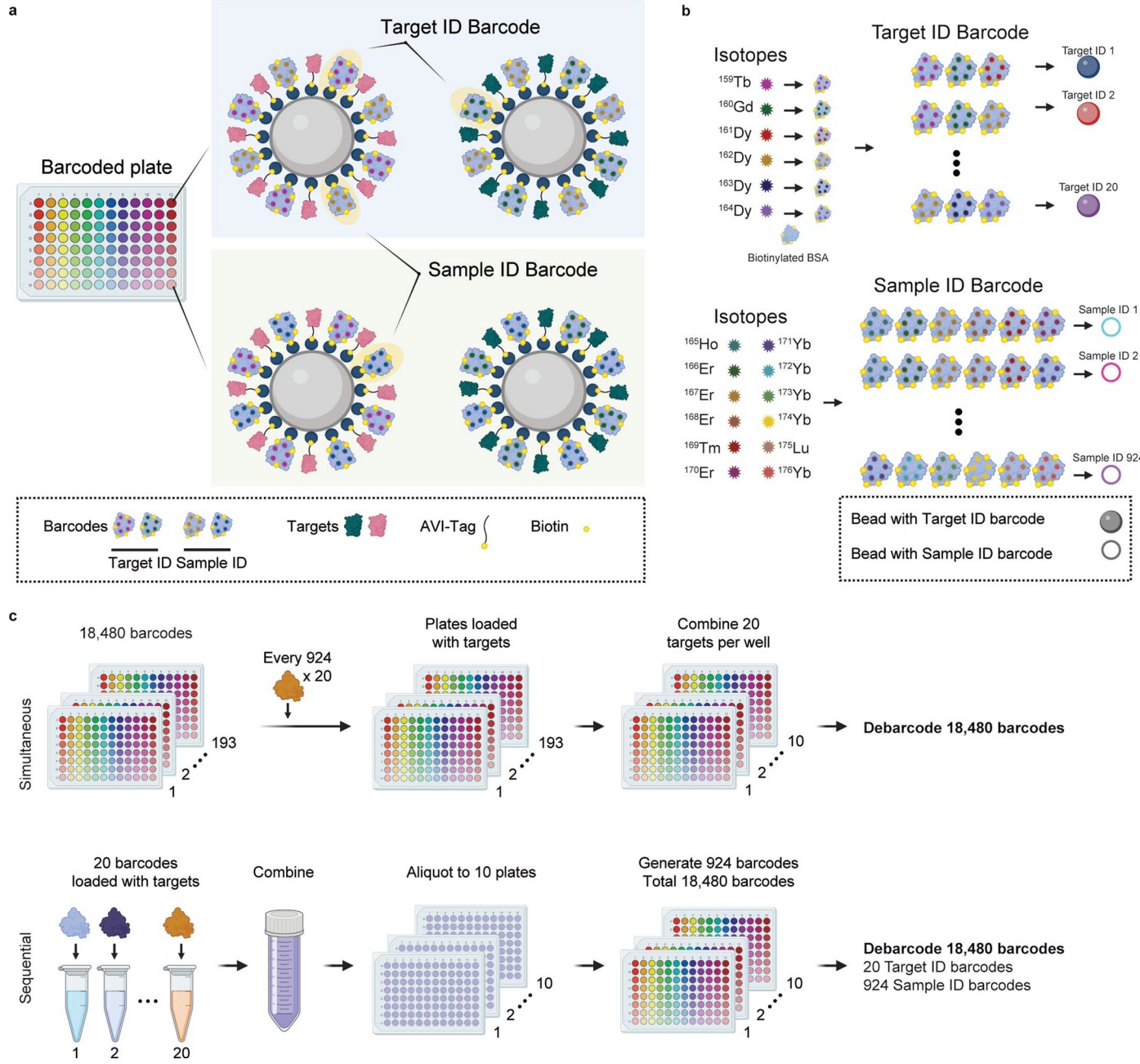

**Extended Data Fig. 1 | Schematic of isotope loading for the double barcoding strategy. a**, Illustration of a 96-well plate of beads used in the double barcoding strategy. Each well contains beads barcoded with 20 target ID and one sample ID. **b**, Schematic of barcode combinations loaded on beads. A six-choose-three barcoding strategy using isotopes $^{159}$Tb to $^{164}$Dy was used to generate 20 target ID barcodes. A twelve-choose-three barcoding strategy using isotopes $^{165}$Ho to $^{176}$Yb was used to generate 924 sample ID barcodes. **c**, Schematic comparison of simultaneous and sequential isotope loading for the generation of 18,480

barcodes. For the simultaneous barcode generation, 200 plates must be individually barcoded and every 924 wells loaded with the target of interest and lastly combining to 10 plates to run all targets for one sample. For the sequential barcode generation used in this study, 20 isotope combinations are loaded together with their target. The beads were pooled and split to 10 plates. Subsequently, 924 isotope combinations were loaded onto beads in wells of those 10 plates generating a total of 18,480 barcode combinations.

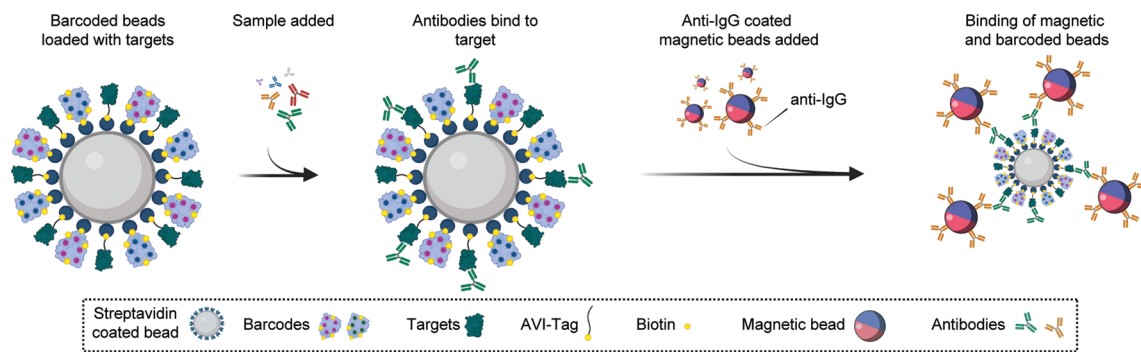

**Extended Data Fig. 2 | Schematic representation of the binding of magnetic and barcoded beads.** Barcoded beads are incubated with serum to allow binding of serum antibodies to targets on the bead. The beads are washed, and then magnetic anti-IgG beads are added that bind to the antibodies present on the barcoded beads.

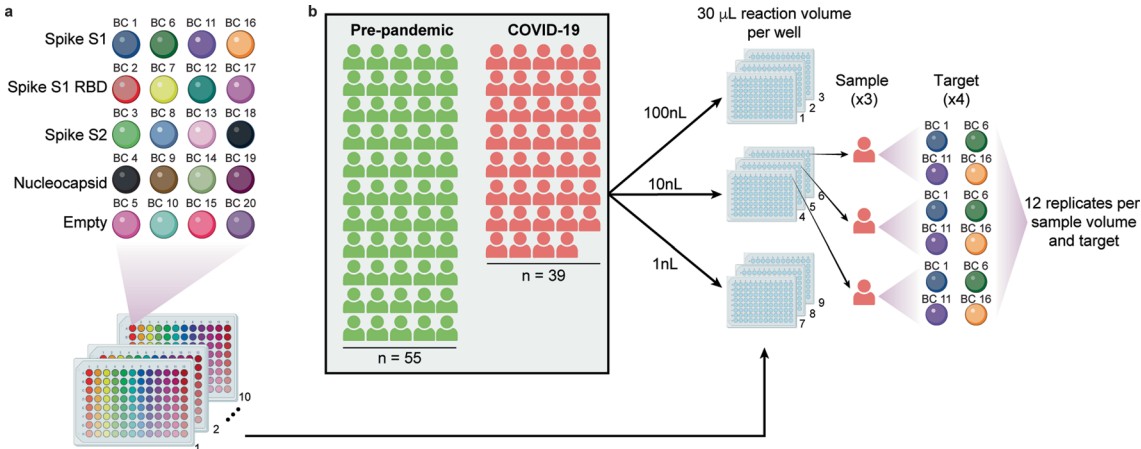

**Extended Data Fig. 3 | Schematic representation of the two-bead assay performed on 55 pre-pandemic and 39 COVID-19 samples. a**, Each well of 10 plates with 924 sample IDs contained 20 Target ID barcodes that were utilized to recognize beads loaded with SARS-CoV-2 Spike S1, Spike S1 RBD, Spike S2, Nucleocapsid, or non-loaded negative control ("Empty"). Each target had four barcodes in this experiment. **b**, For each serum sample, 100 nL, 10 nL, and 1 nL were tested. Each dilution was done in triplicate, and for each replicate each target was tested four times for 12 replicates per sample. Pre-pandemic samples (negative control) are shown in green and COVID-19-positive samples are shown in red.

# Reporting Summary

## Statistics

For all statistical analyses, confirm that the following items are present in the figure legend, table legend, main text, or Methods section.

| n/a | Confirmed | |
|---|---|---|
| ☐ | ☒ | The exact sample size (*n*) for each experimental group/condition, given as a discrete number and unit of measurement |
| ☐ | ☒ | A statement on whether measurements were taken from distinct samples or whether the same sample was measured repeatedly |
| ☒ | ☐ | The statistical test(s) used AND whether they are one- or two-sided<br>*Only common tests should be described solely by name; describe more complex techniques in the Methods section.* |
| ☒ | ☐ | A description of all covariates tested |
| ☒ | ☐ | A description of any assumptions or corrections, such as tests of normality and adjustment for multiple comparisons |
| ☐ | ☒ | A full description of the statistical parameters including central tendency (e.g. means) or other basic estimates (e.g. regression coefficient) AND variation (e.g. standard deviation) or associated estimates of uncertainty (e.g. confidence intervals) |
| ☒ | ☐ | For null hypothesis testing, the test statistic (e.g. *F*, *t*, *r*) with confidence intervals, effect sizes, degrees of freedom and *P* value noted<br>*Give P values as exact values whenever suitable.* |
| ☒ | ☐ | For Bayesian analysis, information on the choice of priors and Markov chain Monte Carlo settings |
| ☒ | ☐ | For hierarchical and complex designs, identification of the appropriate level for tests and full reporting of outcomes |
| ☒ | ☐ | Estimates of effect sizes (e.g. Cohen's *d*, Pearson's *r*), indicating how they were calculated |

*Our web collection on statistics for biologists contains articles on many of the points above.*

## Software and code

Policy information about availability of computer code

| | |
|---|---|
| Data collection | For mass-cytometry experiments, the software used to collect the data was DVS Sciences Cytof instrument control software from Fluidigm. |
| Data analysis | Beads were debarcoded using a custom R script. Data were analysed with R (3.6.2), and plots were created using the ggplot2 R package (3.4.3). All figures were prepared using Adobe Illustrator (24.2.3). |

For manuscripts utilizing custom algorithms or software that are central to the research but not yet described in published literature, software must be made available to editors and reviewers. We strongly encourage code deposition in a community repository (e.g. GitHub). See the Nature Portfolio guidelines for submitting code & software for further information.

## Data

Policy information about availability of data

All manuscripts must include a data availability statement. This statement should provide the following information, where applicable:
- Accession codes, unique identifiers, or web links for publicly available datasets
- A description of any restrictions on data availability
- For clinical datasets or third party data, please ensure that the statement adheres to our policy

The data supporting the findings of the study are provided within the paper and its supplementary information. Raw data related to this manuscript can be accessed at https://zenodo.org/records/10822264.

# Research involving human participants, their data, or biological material

Policy information about studies with <u>human participants or human data</u>. See also policy information about <u>sex, gender (identity/presentation), and sexual orientation</u> and <u>race, ethnicity and racism</u>.

| | |
|---|---|
| Reporting on sex and gender | Sex and gender information for the FIND COVID study was determined on the basis of self-reporting, and reported in Supplementary table 2 as de-identified data at the individual level. Sex and gender information was not collected for the other samples. No studies based on sex and gender were performed in this work. |
| Reporting on race, ethnicity, or other socially relevant groupings | Ethnicity information for the FIND COVID study was determined on the basis of self-reporting, and reported in Supplementary table 2 as de-identified data at the individual level. Ethnicity information was not collected for the other samples. No studies based on ethnicity were performed in this work. |
| Population characteristics | Age, weight, height, schooling level, household information, COVID-19 diagnosis by PCR, and vaccination status for the FIND COVID study were determined on the basis of self-reporting, and are provided in Supplementary table 2 as de-identified data at the individual level. |
| Recruitment | Some samples were collected from COVID-19 convalescent plasma donors who donated plasma at the Stanford Blood Center between April 2020 and May 2020. Other samples were from the FIND COVID study, a CDC-funded longitudinal cohort of individuals recently diagnosed with SARS-CoV-2 infections in the San Francisco Bay Area, initiated in August 2020. |
| Ethics oversight | The Stanford samples study was approved by the Stanford University Institutional Review Board (Protocols IRB-13952, IRB-48973, and IRB-55689). The FIND COVID study was approved by the UCSF Institutional Review Board (Protocol IRB# 20–30388) and given a designation of public health surveillance according to federal regulations as summarized in 45 CFR 46.102(d)(1)(2). |

Note that full information on the approval of the study protocol must also be provided in the manuscript.

# Field-specific reporting

Please select the one below that is the best fit for your research. If you are not sure, read the appropriate sections before making your selection.

☒ Life sciences ☐ Behavioural & social sciences ☐ Ecological, evolutionary & environmental sciences

For a reference copy of the document with all sections, see <u>nature.com/documents/nr-reporting-summary-flat.pdf</u>

# Life sciences study design

All studies must disclose on these points even when the disclosure is negative.

| | |
|---|---|
| Sample size | Most experiments were aimed at describing a new technology; thus, no sample-size calculation was performed. Sample sizes were selected to properly demonstrate and validate technical performance. For the experiments related to the FIND COVID study, the sample size of 542 is higher than those in comparable work. |
| Data exclusions | No data were excluded. |
| Replication | All replication attempts were successful. |
| Randomization | Samples were randomly distributed into 96-well plates. |
| Blinding | Blinding was not necessary because the samples were obtained de-identified. |

# Reporting for specific materials, systems and methods

We require information from authors about some types of materials, experimental systems and methods used in many studies. Here, indicate whether each material, system or method listed is relevant to your study. If you are not sure if a list item applies to your research, read the appropriate section before selecting a response.

## Materials & experimental systems

| n/a | Involved in the study |
|-----|----------------------|
| ☐ | ☒ Antibodies |
| ☒ | ☐ Eukaryotic cell lines |
| ☒ | ☐ Palaeontology and archaeology |
| ☒ | ☐ Animals and other organisms |
| ☒ | ☐ Clinical data |
| ☒ | ☐ Dual use research of concern |
| ☒ | ☐ Plants |

## Methods

| n/a | Involved in the study |
|-----|----------------------|
| ☒ | ☐ ChIP-seq |
| ☒ | ☐ Flow cytometry |
| ☒ | ☐ MRI-based neuroimaging |

## Antibodies

| | |
|---|---|
| Antibodies used | Information of the antibodies used in this study is also detailed in Methods:<br>human anti-Spike S1 (Invivogen, #srbd-mab1)<br>MACS anti-IgG microbeads (Miltenyi Biotec, #130-047-501)<br>anti-IgG-alexa647 (Thermo Fischer Scientific, #A21445)<br>gold anti-IgG antibody (Nanoprobes, #2053)<br>anti-IgM antibody (BioLegend, #314502)<br>horseradish peroxidase-conjugated goat anti-human IgG (Thermo Fisher, #62-8420) |
| Validation | Antibody–antigen interaction experimentally confirmed in vitro. |

