## [Peer Review File · Nature Biomedical Engineering]

High-throughput multiplexed serology via the mass-spectrometric analysis of isotopically barcoded beads

Corresponding author: Xavier Rovira-Clavé

Editorial note

This document includes relevant written communications between the manuscript's corresponding author and the editor and reviewers of the manuscript during peer review. It includes decision letters relaying any editorial points and peer-review reports, and the authors' replies to these (under 'Rebuttal' headings). The editorial decisions are signed by the manuscript's handling editor, yet the editorial team and ultimately the journal's Chief Editor share responsibility for all decisions.

Any relevant documents attached to the decision letters are referred to as **Appendix #**, and can be found appended to this document. Any information deemed confidential has been redacted or removed. Earlier versions of the manuscript are not published, yet the originally submitted version may be available as a preprint. Because of editorial edits and changes during peer review, the published title of the paper and the title mentioned in below correspondence may differ.

Correspondence

Thu 18 Jan 2024

Decision on Article nBME-23-2219

Dear Dr Rovira Clave,

Thank you again for submitting to *Nature Biomedical Engineering* your manuscript, "Highly scalable multiplex serology by mass spectrometry analysis of beads". The manuscript has been seen by 4 experts; however, despite our chasing efforts one reviewer has yet to provide a report (should the reviewer provide it, we will send it to you in due course). You will find the reports of 3 reviewers at the end of this message, and please excuse the unusual delay in reaching a decision

You will see that the reviewers appreciate the work. However, they express concerns about the degree of support for the claims, and provide useful suggestions for improvement. We hope that with significant further work you can address the criticisms and convince the reviewers of the merits of the study. In particular, we would expect that a revised version of the manuscript provides:

* Improved methodological reporting and clarification of all necessary information, as per the relevant comments of reviewers. In particular, I suggest you post a protocol in [protocolexchange](https://protocolexchange.researchsquare.com) (<https://protocolexchange.researchsquare.com>) before you submit the revision.

When you are ready to resubmit your manuscript, please upload the revised files, a point-by-point rebuttal to the comments from all reviewers, the reporting summary, and a cover letter that explains the main improvements included in the revision and responds to any points highlighted in this decision.

Please follow the following recommendations:

* Clearly highlight any amendments to the text and figures to help the reviewers and editors find and understand the changes (yet keep in mind that excessive marking can hinder readability).- * If you and your co-authors disagree with a criticism, provide the arguments to the reviewer (optionally, indicate the relevant points in the cover letter).
- * If a criticism or suggestion is not addressed, please indicate so in the rebuttal to the reviewer comments and explain the reason(s).
- * Consider including responses to any criticisms raised by more than one reviewer at the beginning of the rebuttal, in a section addressed to all reviewers.
- * The rebuttal should include the reviewer comments in point-by-point format (please note that we provide all reviewers will the reports as they appear at the end of this message).
- * Provide the rebuttal to the reviewer comments and the cover letter as separate files.

We hope that you will be able to resubmit the manuscript within 20 weeks from the receipt of this message. If this is the case, you will be protected against potential scooping. Otherwise, we will be happy to consider a revised manuscript as long as the significance of the work is not compromised by work published elsewhere or accepted for publication at *Nature Biomedical Engineering*.

We hope that you will find the referee reports helpful when revising the work. Please do not hesitate to contact me should you have any questions.

Best wishes,

Liqian

Dr Liqian Wang
Associate Editor, Nature Biomedical Engineering

Reviewer #1 (Report for the authors (Required)):

see review attached

Appendix 1

Reviewer #2 (Report for the authors (Required)):

The authors demonstrate the capabilities of multiplexing serology measurements using mass cytometry. Multiplexing is realized by the utilization of numerous uniquely stable isotopically labelled beads, which can be loaded with multiple protein targets or antigens. The authors showed that polystyrene beads can be robustly loaded with distinct stable isotopes and can be reliably quantified via mass cytometry. The technology has been applied to detect antibody levels against multiple SARS-CoV-2 proteins in several clinical plasma and serum samples in a one-tube assay.

Major points. The technique is super interesting, but it has a main limitation. While the barcoding strategy easily allows to generate thousands to tens of thousands of barcodes, one still needs to purify each antigen. Purifying of proteins doesn't scale so easily. So its not easy to exploit the potential of the barcoding in the end.

Line 233ff: Patient specific variability of SarsCOV2 antibody profiles would need independent validation – just getting similar results in replicates does not prove precision. Such a validation would also give confidence in the method.

Technical points

Line 103 and 111: How do you make sure that labelling is uniform across the beads? And how is "uniform labelling" defined? Is it uniform in terms of equally loaded on the bead surface or the amount of 162Dy?

Minor points

Line 119: Why 9 populations? Maybe it is explained in the Suppl Fig 2. What is the importance to have distinct amounts of biotinylated protein carriers? and is it checked?

Line 132: Is it intensity or peak area? How many replicates were analysed? How robust is the bead labelling?

Line 134: I find it difficult to follow how beads are generated. Maybe it makes sense, if you read the additional information in the Supporting Information.

Line 174: The combination of three out of six isotopes generates 20 BC?

Line 229: How is it quantified? And what is quantified? The antibodies? Is it based on counting antibody specific beads?

Line 253: What is the exact measurement?

Line 267: It is stated that a secondary antibody was validated to be used for the quantification of host immunoglobulins but is not explained on what the quantification is based. Are they conjugated with fluorophores?

Reviewer #4 (Report for the authors (Required)):

Drainas et al. report on a new, mass cytometry/spectrometry-based method for the highly multiplexed, high-throughput profiling of antibodies in serum samples. The method takes advantage of the stable isotope barcoding principles enabled by mass cytometry analysis. In the present work, selected isotopes are used to "encode" the sample IDs, while others reflect the identity of the target proteins. In the context of serum analysis, this is experimentally achieved by loading isotope-coded biotinylated proteins on small streptavidin-coated polystyrene beads, which is demonstrated in the first part of the manuscript. A high degree of multiplexing (18,840 unique barcodes in this work) is achieved by double barcoding, which can be demultiplexed computationally based on the unique on-off signature of certain isotopes detected by mass cytometry, resulting in a kind of binary code.

After introducing the labeling strategy and the ability for high-degree multiplexing, the authors demonstrate the application of the concept by (1) detecting IgG antibodies against the SARS-CoV-2 spike protein S1 subunit in plasma from convalescent COVID-19 patients and negative controls (20 samples per group); (2) performing the same assay with a second cohort (39 + 55 samples) with additional dilution steps to prove the sensitivity - for some patient samples, a response could be recorded from as little as 1 nl of sample; and (3) a large-scale application to confirm the levels of IgGs and IgMs against 19 SARS-CoV-2 protein variants in 924 serum samples from a longitudinal COVID study. For the latter application, the method was benchmarked against a commercial IgG high-throughput assay approved for clinical testing. Compared to this reference assay, the CyTOF-based approach allows the processing of all barcoded samples in one tube and provides a vastly extended pool of targets that can be measured simultaneously.

Altogether, the results demonstrate that the method could be successfully implemented for the analysis of real-life clinical samples, is scalable to a highly multiplexed design, and sufficiently sensitive to detect antibody levels in sub- μ l volumes. This points to a promising new direction for mass cytometry/spectrometry-based serological profiling with potential applications in various diagnostic scenarios.

I consider this manuscript an exciting new direction for mass cytometry-based assays and have no major concerns about the technical aspects of the work.

Minor technical criticisms or questions

1. One might argue whether putting the 1 nl sample volume in the abstract is relevant, if very few datapoints can actually be acquired from this volume. Referring to 10 nl is probably also sufficient.
2. The authors repeatedly state that their new method is "low-cost" or "cost-effective", although no details are given. The authors should discuss this a bit and compare the mass cytometry-based approach to other strategies currently used in diagnostic labs.

Missing or unclear details about statistics, protocols or materials

1. In the "Methods" section, no details about flow cytometry and mass cytometry methods are provided.
2. The data set in Mendeley Data linked in the "Data availability" section is not public, and no access details have been provided as far as I am aware.

Stylistic issues

1. The sentence in lines 108-111 needs to be revised.
2. As is frequently the case, the manuscript comes with nicely designed, but somewhat overloaded figures. The text for some panels is impossible to decipher even when enlarging (e.g., Figure 3f, especially the legend for the color scheme). Therefore, the font size must be increased to improve legibility, or the content needs to be split into several figures.

Fri 09 Aug 2024

Decision on Article NBME-23-2219A

Dear Dr Rovira Clave,

Thank you for your patience in waiting for the feedback on your revised manuscript, "Highly scalable multiplex serology by mass spectrometry analysis of beads". Having consulted with the original reviewers (whose comments you will find at the end of this message), I am pleased to write that we shall be happy to publish the manuscript in *Nature Biomedical Engineering*, provided that the points specified in the attached instructions file are addressed.

When you are ready to submit the final version of your manuscript, please upload the files specified in the instructions file.

We encourage authors to take up transparent peer review. If you are eligible and opt in to transparent peer review, we will publish, as a single supplementary file, all the reviewer comments for all the versions of the manuscript, your rebuttal letters, and the editorial decision letters. **If you opt in to transparent peer review, in the attached file please tick the box 'I wish to participate in transparent peer review'; if you prefer not to, please tick 'I do NOT wish to participate in transparent peer review'**. In the interest of confidentiality, we allow redactions to the rebuttal letters and to the reviewer comments. If you are concerned about the release of confidential data, please indicate what specific information you would like to have removed; we cannot incorporate redactions for any other reasons. More information on transparent peer review is available.

Best wishes,

Pep

Pep Pàmies
Chief Editor, Nature Biomedical Engineering

P.S. Nature Portfolio journals encourage authors to share their step-by-step experimental protocols on a protocol-sharing platform of their choice. Nature Portfolio's Protocol Exchange is a free-to-use and open resource for protocols; protocols deposited in Protocol Exchange are citable and can be linked from the published article. More details can be found at www.nature.com/protocolexchange/about.

Reviewer #1 (Report for the authors (Required)):

The Rebuttal provided by the authors is a very well written and interesting document. In the spirit of open review, this would be a document worth including with the publication. It will guide the reader to the very important changes made by the authors in revising their submission.

Reviewer #2 (Report for the authors (Required)):

The Authors have answered all my points sufficiently.

Reviewer #4 (Report for the authors (Required)):

In this revised version, the authors have addressed all my comments appropriately. Specifically, experimental details were added, and the legibility of artwork was substantially improved. The additional Supplementary Notes will help to guide the reader through the somewhat complex workflow.

Nature Biomedical Engineering is a Transformative Journal. Authors may publish their research with us through the traditional subscription access route, or make their paper immediately open access through payment of an article-processing charge. More information about publication options is available.

You may need to take specific actions to comply with funder and institutional open-access mandates. If the work described in the accepted manuscript is supported by a funder that requires immediate open access (as outlined, for example, by Plan S) and your manuscript was originally submitted on or after January 1st 2021, then you will need to select the gold OA route. Authors selecting subscription publication will need to accept our standard licensing terms (including our self-archiving policies), and these will supersede any other terms that the author or any third party may assert apply to any version of the manuscript.

Appendix 1

Highly scalable multiplex serology by mass spectrometry analysis of beads

A. P. Drainas, ..., G. P. Nolan, X. Rovira-Clavé

In this manuscript, the authors report the application of multiple barcodes based on polystyrene microbeads. The authors attach metal-chelating polymers (Maxpar™ reagents) to biotinylated “carrier proteins”, which in turn are attached to streptavidin-coated polystyrene microbeads. The Maxpar™ reagents are labeled with 38 different lanthanide isotopes, ranging in atomic mass from Lanthanum 139 to Ytterbium 176. With various combinations of these labeled polymers, the authors were able to create 18,840 unique isotopically barcoded beads.

They then employed barcoded beads in two types of serological assays. In one set of assays, they attached a biotinylated SARS-CoV-2 spike protein to 924 different beads and tested them against serum samples from a cohort of patients diagnosed with SARS-CoV-2 as well a set of serum samples taken before the SARS-CoV-2 epidemic. In a second set of experiments, they described the simultaneous analysis of the levels of IgG and IgM against 19 proteins in 924 samples. These results are really impressive.

However, the manuscript is not suitable for acceptance in its present form.

The function of the scientific literature is to provide sufficient details so that the work can be reproduced by others. This is a serious shortcoming here. I list below some of the many questions and issues that have to be addressed before the work can be published.

1. What is the carrier protein? How was it chosen and prepared? How was the attachment of the metal-chelating polymers optimized and quantified? How was the stoichiometry of biotin to streptavidin in the multiple coding steps controlled and optimized?
2. The authors claim that the signal of each of the various lanthanide isotopes was identical, yet the transmission factor for each isotope in ICP-MS varies over the range of m/z values. How was this incorporated into reagent design?
3. The authors employed Spherotech, 455 #SVP-30-5 beads, for which they report a diameter variation of 3 to 3.4 μm . This 13% variation in diameter corresponds to a 28% variation in surface area. How does that affect the number of streptavidin molecules per bead, and how did the authors compensate for this variation?
4. There is no mention of non-specific binding, which is always a problem in these types of assays. Where was non-specific binding observed and what did the authors do to minimize non-specific binding in these experiments.
5. The authors used secondary antibodies in the simultaneous analysis of the levels of IgG and IgM. In this type of assay, the secondary antibody has to be labeled with a reporter group. For this purpose, they used gold anti-IgG antibody (Nanoprobes, #2053) to detect IgG and anti-IgM antibody (BioLegend, #314502) to detect IgM. However, the anti-IgM antibody does not contain a reporter group. What did the authors actually do?

The results of this submission are noteworthy, and I would be happy to review a revised submission in which the missing details are provided.

Rebuttal 1

Response to Reviewers

Highly scalable multiplex serology by mass spectrometry analysis of beads

We are grateful to the Editor and Reviewers for assessing this work and for their advice and positive feedback.

In the revised version of the manuscript, we have:

- Generated a step-by-step illustration of the process of bead generation and assay.
- Provided a detailed description of the carrier protein.
- Described the bead properties: the size variation, the surface area, and the capacity to load biotinylated BSA.
- Described the non-specific binding of antibodies to beads and antigens.
- Discussed the assay's scalability, which surpasses antigen production and purification, and highlighted that the substantial current availability of resources can be effectively utilized. We have also included details about the cost-effectiveness of the assay.
- Validated the results of a multiplexed two-bead mass cytometry assay in an independent ELISA assay.
- Included detailed descriptions, replicates, and schematics for various figures and in the methods section of the manuscript to improve the clarity and highlight the robustness of the results. We have also increased font sizes across most figures to improve legibility.
- Included the missing information about one of the two secondary antibodies used.
- Uploaded a detailed protocol to “protocolexchange” and raw data, code, and debarcoding demo to Zenodo.

Below we respond in detail to each of the concerns raised by the Reviewers.

Reviewer #1

In this manuscript, the authors report the application of multiple barcodes based on polystyrene microbeads. The authors attach metal-chelating polymers (Maxpar™ reagents) to biotinylated “carrier proteins”, which in turn are attached to streptavidin-coated polystyrene microbeads. The Maxpar™ reagents are labeled with 38 different lanthanide isotopes, ranging in atomic mass from Lanthanum 139 to Ytterbium 176. With various combinations of these labeled polymers, the authors were able to create 18,840 unique isotopically barcoded beads.

They then employed barcoded beads in two types of serological assays. In one set of assays, they attached a biotinylated SARS-CoV-2 spike protein to 924 different beads and tested them against serum samples from a cohort of patients diagnosed with SARS-CoV-2 as well a set of serum samples taken before the SARS-CoV-2 epidemic. In a second set of experiments, they described the simultaneous analysis of the levels of IgG and IgM against 19 proteins in 924 samples. These results are really impressive.

However, the manuscript is not suitable for acceptance in its present form. The function of the scientific literature is to provide sufficient details so that the work can be reproduced by others. This is a serious shortcoming here. I list below some of the many questions and issues that have to be addressed before the work can be published.

We thank the Reviewer for the positive assessment of our technology and for highlighting that reproducing the procedure might be challenging with the current details. To facilitate the reproducibility of our work by others, we have included **Supplementary Note 2** (page 21, line 621), which describes the assay in a step-by-step fashion. We also address the concerns raised below.

1. What is the carrier protein? How was it chosen and prepared? How was the attachment of the metal-chelating polymers optimized and quantified? How was the stoichiometry of biotin to streptavidin in the multiple coding steps controlled and optimized?

We thank the Reviewer for these questions.

- The carrier protein utilized in this study is Bovine Serum Albumin (BSA).
- BSA was chosen because it is commercially available in a biotinylated format, it is cost-effective, and it has several cysteines for polymer conjugation. Isotope-conjugated BSA was prepared using a standard conjugation protocol, previously described in the materials and methods section named “Conjugation of carrier proteins to isotopes”. In the revised version of the manuscript (page 17, line 473), this section is named “Conjugation of Bovine Serum Albumin (BSA) and antibodies to isotopes”. In this section, we now state: “Biotinylated BSA (Sigma, #A8549) was used as carrier protein”. We have also included the use of BSA in **Figure 1a** and changed the Results section of the revised manuscript (page 3, line 109) to read: “We conjugated the stable isotope Dysprosium 162 (¹⁶²Dy) to biotinylated bovine serum albumin (BSA) and loaded it to streptavidin-coated beads.”.
- The conjugation protocol used in this study is the same protocol that is commonly used to conjugate isotopes to antibodies for mass cytometry, and it did not require optimization (Han, et al., 2018). Isotope conjugation is validated and quantified after every conjugation experiment by loading BSA into beads and analyzing them by mass cytometry. We have included the following sentence in the Materials and Methods section of the revised manuscript (page 17, line 480): “Isotope conjugation was confirmed after every conjugation experiment by loading isotope conjugated BSA into beads and analyzing them by mass cytometry”.

- The number of biotin molecules conjugated to each BSA molecule is variable, the loading of isotopes in each BSA is variable, and the number of streptavidin molecules per bead is variable, therefore, it is challenging to control and optimize this stoichiometry at the bead level. The assumptions taken and the specific numbers to control and optimize the stoichiometry at the population level are as follows:
 - In 1 mL of 0.5% w/v particles there are 5 mg of particles.
 - Each mg of a 4 μm particle typically binds 5-10 μg of biotinylated antibody (per manufacturer specs).
 - The surface area of 1 mg of 4 μm particles is 14.29 cm^2 . The surface area of 1 mg of 3 μm particles is 19.05 cm^2 .
 - The number of beads in 1 mg of 4 μm particles is 0.028×10^9 . The number of beads in 10 mg of 3 μm particles is 0.067×10^9 .
 - Thus, the surface of a 4 μm particle is 1.79 larger than the surface of a 3 μm particle, but the surface of 1 mg of 3 μm particles is 1.33 larger than the surface of 1 mg of 4 μm particles.
 - Spherotech # SVP-30-5 beads have a size range of 3 to 3.4 μm diameter.
 - Each mg of a 3 μm particle typically binds 6.67-13.33 μg of biotinylated antibody, and each mg of a 3.4 μm particle typically binds 5.88-11.76 μg of biotinylated antibody.
 - Altogether, 1 mL (5 mg) of Spherotech # SVP-30-5 beads can bind 29.41-66.67 μg of biotinylated antibody.
 - A typical antibody contains 80 lysines, most of them exposed for biotinylation. BSA contains 60 lysines, most of them exposed for biotinylation. The biotinylation process can be highly variable depending on the conditions.
 - All considered, we assumed that 1 mL of Spherotech #SVP-30-5 beads can bind at least 29.41-66.67 μg of biotinylated BSA.

We performed two sequential loading steps of 1 mL of Spherotech # SVP-30-5 beads in a 1 mL volume. The first one with 3 $\mu\text{g}/\text{mL}$ of biotinylated BSA and the second one with 6 $\mu\text{g}/\text{mL}$ of biotinylated BSA. We hypothesized that this sequential loading strategy would be compatible with the binding capability of Spherotech #SVP-30-5 beads. The results described in **Figure 2** and **Supplementary Figures 3 to 12** demonstrate that the loading isotopes is enough to successfully identify all barcodes at the population level. We did not perform further refinement because this extensive validation demonstrate that the proposed workflow enables the successful identification of all expected barcodes. In the revised version of the manuscript, we have included the **Supplementary Note 3** (page 21, line 621), which describes the numbers to control and optimize the stoichiometry of biotin to streptavidin at the population level.

2. The authors claim that the signal of each of the various lanthanide isotopes was identical, yet the transmission factor for each isotope in ICP-MS varies over the range of m/z values. How was this incorporated into reagent design?

The Reviewer is right that the transmission factor for each isotope in ICP-MS varies over the range of m/z values (Tricot, et al., 2015) (**Figure R1**) and that this phenomenon applies to the serology assay by mass cytometry described in this manuscript. To generate

Figure R1. Ratios of observed to expected counts for isotopes ranging from ^{139}La to ^{176}Yb . Adapted from *Tricot, et al., 2015*.

the 18,480 barcodes, we used the isotopes ranging from ^{159}Tb to ^{176}Yb , all of them with a similar transmission factor (**Figure R1**).

Importantly, the relevant feature for the assay is assessing the presence or absence of an isotope in a single bead, rather than how the signal of an isotope compares to the signal of another isotope in the same bead. For example, in a hypothetical bead loaded with the isotopes ^{143}Nd and ^{174}Yb , it would not affect classification if the signal of ^{174}Yb was 3 times higher than the signal of ^{143}Nd . The hypothetical bead would be classified as having ^{143}Nd and ^{174}Yb .

Supplementary Figures 7 and 9 show that the recorded isotope intensities are enough to assess the presence or absence of an isotope in beads, independently of the transmission factor of the isotopes.

In the previous version of the manuscript, we wrote:

“the intensities for each isotope were comparable (**Fig. 1e**)”.

We agree with the Reviewer that the wording might be misleading. In the revised manuscript, we have substituted this sentence (page 3, line 117) for:

“the intensities for each isotope were similar despite the distinct transmission factors for each isotope typically observed in ICP-MS (**Fig. 1e** and **Supplementary Fig. 1d**).”

The Discussion section of the revised version of the manuscript (page 14, line 387), now reads:

“The remaining isotopes could be harnessed to increase the number of barcodes, but careful validation should be performed given the distinct transmission factors for each isotope in ICP-MS.”

3. The authors employed Spherotech, 455 #SVP-30-5 beads, for which they report a diameter variation of 3 to 3.4 μm . This 13% variation in diameter corresponds to a 28% variation in surface area. How does that affect the number of streptavidin molecules per bead, and how did the authors compensate for this variation?

We thank the Reviewer for pointing this out. The Reviewer is right that the Spherotech # SVP-30-5 beads have a diameter variation of 3 to 3.4 μm . The larger the diameter, the higher the number of streptavidin molecules on the bead surface. The larger the diameter, the lower the number of beads for the same weight of polymer. Based on the information provided in Comment #1 and the new **Supplementary Note 3** (page 21, line 621):

- We assume that 1 mL of 3 μm particles can bind 33-67 μg of biotinylated BSA.
- We assume that 1 mL of 3.4 μm particles can bind 29-59 μg of biotinylated BSA.

We compensated for this variation by acquiring multiple beads for each barcode and calculating the median values of the isotope intensities for each bead population. In **Supplementary Figure 12**, we show that in a typical run we collect 25 to 100 single beads for each of the 18,480 barcodes. This is enough to collect beads of a variety of sizes for each barcode.

We have included the following sentence in the Discussion section of the revised manuscript (page 14, line 396):

“The beads used in this study have a diameter variation of 3 to 3.4 μm , with larger beads having a higher number of streptavidin molecules on their surface. The use of beads with a narrower range of diameter variation will also improve the throughput of the assay by reducing the number of required beads for analysis.”

4. There is no mention of non-specific binding, which is always a problem in these types of assays. Where was non-specific binding observed and what did the authors do to minimize non-specific binding in these experiments.

We thank the Reviewer for this comment. Several factors contribute to minimize non-specific binding in our assay:

- The nanoparticles are coated with BSA. This coating acts as isotope carrier for bead debarcoding but also as a blocker of sites on the surface that might be prone to non-specific binding.
- We use low sample volumes, typically 1 μL of sample or less in a 40 μL reaction volume.
- The buffer contains 0.5% BSA as protein blocking additive and 0.001% Tween-20 as a non-ionic surfactant.
- Each sample is incubated with beads without antigen. These beads are used as internal normalization reagents to account for potential non-specific binding.

We assessed non-specific binding in **Supplementary Figure 21** of the manuscript, which is included below as **Figure R2** for convenience of the Reviewer. In this experiment we assessed non-specific binding to SARS-CoV-2 Spike S1 loaded beads of pre-pandemic samples and the secondary anti-IgG antibody. There were 4 wells with beads without any plasma added, 8 wells with negative control samples, and 8 wells with 1 positive sample titrated from 0.0625 μL to 8 μL of plasma (**Figure R2a**). The beads were incubated with plasma and then thoroughly washed and incubated with a secondary anti-IgG ^{197}Au (gold) conjugated antibody (**Figure R2a**). The samples were washed, combined in one tube, and run in the mass cytometer. The intensity of the ^{197}Au is shown in the histograms (**Figure R2b**). Samples 1-4 showed low ^{197}Au signal, suggesting minimal non-specific binding of anti-IgG to the beads. Samples 5-12, the pre-pandemic samples, displayed a slightly higher ^{197}Au signal than samples 1-4, indicating potential non-specific binding of the IgGs to the beads. The titrated positive sample displayed signal higher than the negative controls. In summary, non-specific binding is observed in pre-pandemic samples, but the signal is lower than the observed in positive samples.

Figure R2. Validation of secondary antibodies for the quantification of immunoglobulins on isotope-barcoded beads. (a) Schematic representation of the secondary antibody validation experiment. (b) Heatmap and histograms of IgG levels on Spike S1-loaded, isotope-barcoded beads incubated with plasma samples from one COVID-19 positive sample and eight plasma samples collected prior to the COVID-19 pandemic (1 μL each). IgG levels on beads were quantified by mass cytometry using anti-human IgG conjugated to gold nanoparticles.

We have updated **Supplementary Fig. 21** to include the schematic representation of the experiment and modified the Results section of the revised manuscript (page 10, line 286), to read:

“We observed minimal non-specific binding of the secondary anti-IgG antibody to beads (**Supplementary Fig. 21**, Samples 1-4) and low non-specific binding of pre-pandemic samples (**Supplementary Fig. 21**, Samples 5-12).”

5. The authors used secondary antibodies in the simultaneous analysis of the levels of IgG and IgM. In this type of assay, the secondary antibody has to be labeled with a reporter group. For this purpose, they used gold anti-IgG antibody (Nanoprobes, #2053) to detect IgG and anti-IgM antibody (BioLegend, #314502) to detect IgM. However, the anti-IgM antibody does not contain a reporter group. What did the authors actually do?

We apologize for this oversight. Anti-IgM was conjugated to ^{115}In , using commercially available tools. We have included the information in the Materials and Methods sections of the revised version of the manuscript.

In page 17, line 481:

“Anti-human IgM was conjugated to ^{115}In chelated Maxpar X8 polymers following the same procedure.”

and in page 21, line 610:

“The beads were stained with 25 mL CSM-T containing 256 ng/L gold anti-IgG antibody (Nanoprobes, #2053) and 256 ng/L ^{115}In -conjugated anti-IgM antibody (BioLegend, #314502) for 30 minutes.”

The results of this submission are noteworthy, and I would be happy to review a revised submission in which the missing details are provided.

We are grateful to the Reviewer for the insightful assessment of the manuscript, and we hope the details we have included in this revision will help others wanting to reproduce the work.

Reviewer #2

The authors demonstrate the capabilities of multiplexing serology measurements using mass cytometry. Multiplexing is realized by the utilization of numerous uniquely stable isotopically labelled beads, which can be loaded with multiple protein targets or antigens. The authors showed that polystyrene beads can be robustly loaded with distinct stable isotopes and can be reliably quantified via mass cytometry. The technology has been applied to detect antibody levels against multiple SARS-CoV-2 proteins in several clinical plasma and serum samples in a one-tube assay.

Major points

The technique is super interesting, but it has a main limitation. While the barcoding strategy easily allows to generate thousands to tens of thousands of barcodes, one still needs to purify each antigen. Purifying of proteins doesn't scale so easily. So its not easy to exploit the potential of the barcoding in the end.

We thank the Reviewer for the kind words.

The Reviewer is absolutely correct that antigen purification is a main limitation of the technique. This is especially true for new antigens (e.g., in the event of a new pandemic threat), which may require weeks or months of production and purification. For already known antigens, there is a large collection of commercially available purified antigens that can be purchased to assemble panels. Validation is required for these antigens, but their off-the-shelf availability can ameliorate the issue.

We believe that future panels of tens of antigens to assess antibody levels in hundreds to thousands of samples will take full advantage of the barcoding capacity of the technology while still being compatible with reasonable validation efforts. As a comparison, technologies such as CyTOF, CITE-seq, CODEX, MIBI, Xenium, and MERFISH require validation of panels of tens or hundreds of probes, and those efforts are routinely performed in academic laboratories.

We have included the following sentences in the Discussion section of the revised manuscript to acknowledge this issue (page 15, line 400):

“Additionally, the assembling of antigen panels is constrained by the scalability of protein purification. Our technology scales faster than the process of integrating new purified proteins into the panel, thus posing a challenge in fully leveraging the multiplexing potential of the technology. The commercial availability of antigens ameliorates the issue, although extensive validation will be required when assembling new antigen panels.”

Line 233ff: Patient specific variability of SarsCOV2 antibody profiles would need independent validation – just getting similar results in replicates does not prove precision. Such a validation would also give confidence in the method.

We thank the Reviewer for this comment. Getting similar results in replicates is a strong indicator of precision, however, we agree with the Reviewer that it does not prove it. To improve the confidence on a new assay, the results must align with a reference value obtained with an already validated technique.

In **Figure 3d** of the originally submitted manuscript, we showed independent validation of the mass cytometry assay by ELISA in 20 plasma samples from COVID-19 convalescent donors.

Additionally, in **Supplementary Fig. 15** we showed independent validation of the mass cytometry assay by flow cytometry analysis in the same 20 plasma samples from COVID-19 convalescent donors, in three independent experiments. The results from the mass cytometry assay align with the results from two independent validated assays (flow cytometry and ELISA), indicating that this single-plex mass cytometry assay is precise.

To clarify that the flow cytometry assay represents an independent validation, we have modified the Results section of the revised manuscript (page 7, line 220), to read:

“Furthermore, we observed similar results when samples were analyzed individually in an independent flow cytometry assay rather than pooled together in the mass cytometry assay (**Supplementary Fig. 15a**).

To further strengthen these results, we have validated the results of a multiplexed mass cytometry assay to the results of an ELISA assay. We assessed the levels of IgGs against SARS-CoV-2 Spike S1, Spike S1 RBD, and nucleocapsid using both assays in the same 20 plasma samples from COVID-19 convalescent donors. The results from the multiplexed mass cytometry results align with the results obtained with ELISA for all three proteins (**Figure R3**), indicating that this multiplexed mass cytometry assay is precise.

Figure R3. Individual and pooled analysis of COVID-19 samples. Dot plot of the ratio of beads in the flow-through to baseline of the two-bead mass cytometry assay (X-axis) versus the ELISA values per sample (Y-axis) for Spike S1, Spike S1 RBD, and Nucleocapsid. Samples were analyzed individually for ELISA and were pooled for mass cytometry analysis. Samples were randomly distributed.

In the revised version of the manuscript, we have included **Figure R3** as a new **Supplementary Fig. 15b** and the following sentence in the Results section (page 7, line 222):

“We additionally observed proper alignment of a multiplexed assessment of the levels of antibodies against SARS-CoV-2 Spike S1, the receptor binding domain of Spike S1 (Spike S1 RBD), and the Nucleocapsid protein to the results of an ELISA assay (**Supplementary Fig. 15b**).”

We have also modified the following sentence in the Results section of the manuscript:

“These results validate the two-bead system for antibody detection in patient plasma samples.”

To read (page 8, line 228):

“These results show that the mass cytometry results align with the results obtained with ELISA and flow cytometry, validating the two-bead system for antibody detection in patient plasma samples.”

We further validated the SARS-CoV-2 antibody profiles showed in **Figure 3e** by testing 500 nL of all the 55 pre-pandemic and 39 COVID-19 samples on independent ELISA assays targeting Spike S1 and Nucleocapsid. We compared the ELISA values to the data from 100 nL and 10 nL of plasma sample used for the two-bead mass cytometry assay and observed a strong correlation between the two assays (**Figure R4**; 10 nL or Spike S1 and 100 nL for Nucleocapsid in the X-axis). Overall, the ELISA experiments validate the observations made using the new mass cytometry assay.

Figure R4. Individual and pooled analysis of COVID-19 samples. Dot plot of the ratio of beads in the flow-through to baseline of the two-bead mass cytometry assay (X-axis) versus the ELISA values per sample (Y-axis) for Spike S1 and Nucleocapsid. Samples were analyzed individually for ELISA and were pooled for mass cytometry analysis. Samples were randomly distributed.

In the revised version of the manuscript, we have included Figure R4 as a new **Supplementary Fig. 20b** and the following sentence in the Results section (page 8, line 251):

“Additionally, the levels of antibodies against SARS-CoV-2 Spike S1 and Nucleocapsid strongly correlated with independent ELISA assays conducted on the same set of samples (**Supplementary Fig. 20b**).“

Technical points

Line 103 and 111: How do you make sure that labelling is uniform across the beads? And how is “uniform labelling” defined? Is it uniform in terms of equally loaded on the bead surface or the amount of ¹⁶²Dy?

We thank the Reviewer for pointing this out. We refer to uniform labeling as single beads in the population having similar isotope intensities (i.e., the mass cytometry readout of the amount of ^XLn per single bead). We do not analyze whether the isotope-conjugated BSA is distributed equally though the surface of the bead because the relevant feature for barcode identification is assessing the presence or absence of isotopes at the single bead level, which can be achieved at high-throughput by mass cytometry.

We ensure uniform labeling across the beads by incubating streptavidin-coated polystyrene beads with a diameter of 3 to 3.4 μm with 2.5 μg/mL isotope-conjugated biotinylated carrier proteins for 30 minutes at room temperature, and subsequently validating the process by mass cytometry analysis.

We have modified the Results section to clarify this concept, and now the revised manuscript reads (page 3, line 105):

“We reasoned that binding of isotope-conjugated biotinylated proteins to streptavidin-coated polystyrene beads would provide a high load of isotopes per bead and a uniform labeling, the latter understood as single beads in the population having similar isotope intensities (Fig. 1a and Supplementary Fig. 1a)”.

Minor points

Line 119: Why 9 populations? Maybe it is explained in the Suppl Fig 2. What is the importance to have distinct amounts of biotinylated protein carriers? and is it checked?

We thank the Reviewer for these questions. A relevant feature for the assay is assessing the presence or absence of an isotope in a single bead. Another relevant feature for the assay is multiplexing (i.e., the readout of multiple isotopes in the same bead). By combining the presence or absence of two isotopes, with two distinct amounts when present, there are 9 distinct possible options:

	Isotope 1: absent	Isotope 1: 1X amounts	Isotope 1: 10X amounts
Isotope 2: absent	1	2	3
Isotope 2: 1X amounts	4	5	6
Isotope 2: 10X amounts	7	8	9

Having distinct amounts is not relevant for the generation of the barcoding strategy described in Figure 2 and used in Figures 3 and 4 of this manuscript, but it highlights the robustness, tunability, and flexibility of the system. The system is robust because it is compatible with a range of amounts of biotinylated protein carriers, it is tunable because we can use distinct concentrations at will, and it is flexible because we can use distinct schemes for isotope loading.

Yes, we checked that the beads have distinct amounts of biotinylated protein carriers by loading the beads with distinct concentrations, subsequently loading them with one additional isotope (ranging from ¹⁵⁹Tb to ¹⁶⁷Er) for bead tracing, and analyzing the pool by mass cytometry, as demonstrated in **Supplementary Figure 2**.

To better explain the experiment performed in **Supplementary Figure 2**, we have included a schematic representation of the workflow in the revised version of the manuscript (**Figure R5**).

Figure R5. Schematic representation of the 9 populations analyzed in the experiment.

Line 132: Is it intensity or peak area? How many replicates were analysed? How robust is the bead labelling?

We thank the Reviewer for these questions. The reported values are the median of the isotope intensities in each bead population. We have modified the manuscript (page 4, line 134) to read:

“Color shows the median of the normalized isotope intensities for odd and even bead populations”.

The odd/even experiment was performed twice. **Figure 1d and e** show a representative experiment, and the second one is now included in **Supplementary Figure 1c and d** of the revised version of the manuscript.

The bead labelling is extremely robust. All bead labelling procedures following the section “**Bead loading with isotope-conjugated carrier proteins**” described in Materials and Methods were successful.

Line 134: I find it difficult to follow how beads are generated. Maybe it makes sense, if you read the additional information in the Supporting Information.

We thank the Reviewer for this comment. We have written **Supplementary Note 2** (page 21, line 621), a document describing the assay in a step-by-step fashion that includes a description of how beads are generated.

Line 174: The combination of three out of six isotopes generates 20 BC?

Yes, the combination of three out of six isotopes generates 20 barcodes. The mathematical formula is:

$${}^n C_r = \frac{n!}{r!(n-r)!}$$

Where n is the number of isotopes and r is the number of isotopes chosen from the pool. The largest value for nCr is obtained at $r=n/2$, when n is an even number. For example, if we have 6 hypothetical isotopes named A, B, C, D, E, and F, and we pick three at a time, the 20 possible combinations are ABC, ABD, ABE, ABF, ACD, ACE, ACF, ADE, ADF, AEF, BCD, BCE, BCF, BDE, BDF, BEF, CDE, CDF, CEF, and DEF.

In the revised version of the manuscript, the legend of **Figure 2** now reads (page 6, line 177):

“Schematic representation of the workflow for the generation of isotope-barcoded beads. For target ID barcoding, beads are labeled with three of the six target ID isotopes (^{159}Tb to ^{164}Dy) to generate 20 barcodes. For sample ID barcoding, beads are labeled with six of the 12 sample ID isotopes (^{165}Ho to ^{176}Yb) to generate 924 barcodes.”

Line 229: How is it quantified? And what is quantified? The antibodies? Is it based on counting antibody specific beads?

We thank the Reviewer for these questions. In the two-bead selection system, we count the number of beads for each barcode in the baseline and the flowthrough. The ratio of the bead count in the flowthrough and the baseline bead mixture provides an estimate of host antibody levels.

The Results section of the revised version of the manuscript now reads (page 7, line 203):

“For each barcode, the ratio of the bead count in the flowthrough and the baseline bead mixture provides an estimate of host antibody levels, with low ratio values implying the presence of host antibodies specific to the antigen in a given sample.”

and also (page 8, line 237):

“Each sample and dilution were quantified 12 times using three sample replicates and four target replicates by calculating the ratio of the bead count in the flowthrough and the baseline bead mixture in each replicate (**Supplementary Fig. 17**).”

Line 253: What is the exact measurement?

We thank the Reviewer for this question, as we now realize that the exact measurement is not well described. The measurement in the x-axis of **Figure 3d** is the ratio between the bead count of the beads in the flowthrough to the bead count of the baseline. We have now updated the x-axis of **Figure 3d** to read: “Flow-through to baseline ratio (ratio of the # of beads in the flow-through to the # of beads in the baseline)”.

We have modified the legend of **Figure 3d** (page 10, line 264), to read:

“Comparison of ELISA (y-axis) versus the two-bead assay (x-axis) in the 20 COVID-19-positive samples shown in panel (c). The two-bead assay result is shown as the ratio of the number of beads in the flowthrough to the number of beads in the baseline.”

Line 267: It is stated that a secondary antibody was validated to be used for the quantification of host immunoglobulins but is not explained on what the quantification is based. Are they conjugated with fluorophores?

Drainas, et al.

We apologize for this oversight. Secondary antibodies were conjugated to isotopes. We used anti-human IgG conjugated to gold (^{197}Au) nanoparticles and anti-human IgM conjugated to ^{115}In -chelated Maxpar X8 polymers.

The Results section of the revised version of the manuscript now reads (page 10, line 277):

“We validated the compatibility of secondary antibodies conjugated to isotopes for the quantification of host immunoglobulins bound to our isotope-barcoded beads using positive and negative controls.”

The Materials and Methods section of the revised version of the manuscript now reads (page 21, line 610):

“The beads were stained with 25 mL CSM-T containing 256 ng/L gold anti-IgG antibody (Nanoprobes, #2053) and 256 ng/L ^{115}In -conjugated anti-IgM antibody (BioLegend, #314502) for 30 minutes.”

Reviewer #4

Drainas et al. report on a new, mass cytometry/spectrometry-based method for the highly multiplexed, high-throughput profiling of antibodies in serum samples. The method takes advantage of the stable isotope barcoding principles enabled by mass cytometry analysis. In the present work, selected isotopes are used to "encode" the sample IDs, while others reflect the identity of the target proteins. In the context of serum analysis, this is experimentally achieved by loading isotope-coded biotinylated proteins on small streptavidin-coated polystyrene beads, which is demonstrated in the first part of the manuscript. A high degree of multiplexing (18,840 unique barcodes in this work) is achieved by double barcoding, which can be demultiplexed computationally based on the unique on-off signature of certain isotopes detected by mass cytometry, resulting in a kind of binary code.

After introducing the labeling strategy and the ability for high-degree multiplexing, the authors demonstrate the application of the concept by (1) detecting IgG antibodies against the SARS-CoV-2 spike protein S1 subunit in plasma from convalescent COVID-19 patients and negative controls (20 samples per group); (2) performing the same assay with a second cohort (39 + 55 samples) with additional dilution steps to prove the sensitivity - for some patient samples, a response could be recorded from as little as 1 nl of sample; and (3) a large-scale application to confirm the levels of IgGs and IgMs against 19 SARS-CoV-2 protein variants in 924 serum samples from a longitudinal COVID study. For the latter application, the method was benchmarked against a commercial IgG high-throughput assay approved for clinical testing. Compared to this reference assay, the CyTOF-based approach allows the processing of all barcoded samples in one tube and provides a vastly extended pool of targets that can be measured simultaneously.

Altogether, the results demonstrate that the method could be successfully implemented for the analysis of real-life clinical samples, is scalable to a highly multiplexed design, and sufficiently sensitive to detect antibody levels in sub- μ l volumes. This points to a promising new direction for mass cytometry/spectrometry-based serological profiling with potential applications in various diagnostic scenarios.

I consider this manuscript an exciting new direction for mass cytometry-based assays and have no major concerns about the technical aspects of the work.

We appreciate the Reviewer's time taken reading our manuscript, and the positive view on the progress it represents for mass cytometry-based assays.

Minor technical criticisms or questions

1. One might argue whether putting the 1 nl sample volume in the abstract is relevant, if very few datapoints can actually be acquired from this volume. Referring to 10 nl is probably also sufficient.

We thank the Reviewer for this comment, and we agree it is better to modify this claim. In the previous version of the abstract, we wrote:

"We demonstrate high-throughput specific detection of antibodies using minimal sample volumes as low as 1 nanoliter."

In the revised abstract, we have substituted this sentence (page 1, line 38) for:

"We demonstrate high-throughput specific detection of antibodies using sample volumes in the hundreds of nanoliter range."

2. The authors repeatedly state that their new method is "low-cost" or "cost-effective", although no details are given. The authors should discuss this a bit and compare the mass cytometry-based approach to other strategies currently used in diagnostic labs.

The Reviewer is right, details on the economics of the assay are missing.

Multiplex serology by mass spectrometry analysis of beads is a cost-effective technology at large number of samples and targets per run. For instance, if we take a conservative estimate, it would be feasible implementing 100 runs per mass cytometer per year (approximately 2 runs of 4 to 6 hours per week). In this scenario, the cost of materials for each run is estimated to be **\$1,908.55 (Figure R6)**. Each run could accommodate 924 samples (including controls), and therefore the cost of materials per sample is estimated to be **\$2.07**. Each sample would be tested for up to 20 targets and therefore the cost of materials is estimated to be **\$0.10 per analyte per sample**. These cost estimates do not include operational and instrumental costs, which would be the capital and annual maintenance costs of a mass cytometer, the annual full-time salaries of one or two technicians, amongst others.

Item	Cost per unit	number of 924 assays per number of samples per	Cost per sample	Units needed per year	Cost per year	Notes
Number of tests per sample	20					
Samples per assay (samples + internal assay QC)	924					
Assays per instrument each year	100					
Isotopes for sample barcode	12					
Isotopes for target barcode	8					
Functionalized Polystyrene beads	\$245.00	5.00	4620	\$0.0530	20	\$4,900.00 https://www.schmidtech.com/usa_pdf_per.htm
96 well liquid handler tips (10 pack)	\$60.00	1.00	924	\$0.0649	100	\$6,000.00 https://www.amazon.com/200ul-Plastic-Tip-Universal/Batches%2F730C88860UQ9089D488F2?ref=aod_aod_3142-0287336
Irradio tips (pack of 50 boxes)	\$450.00	5.00	4620	\$0.0974	20	\$9,000.00 https://www.kalium.com/en/industrial-automated-high-throughput-consumables-for-96-well-plates/broch-96-96-plates.html
96 well plates (50 pack)	\$400.00	2.50	2310	\$0.1732	40	\$18,000.00 https://databio.com/en/for-research/96-well-plates/96-well-plates-96-well-plates/96-well-plates-96-well-plates
Anti-IgG-Gold	\$401.00	10.00	9240	\$0.0434	10	\$4,010.00 https://www.caspr.com/products/antibodies/antibodies-conjugates.html
Lancets	\$13.00	0.11	100	\$0.1300	924	\$12,912.00 https://www.amazon.com/M&K-Esson-16-PBS-180-Safety-Lancet-Needle-09-0117044M01refsr_1_87refr-2130CHF-KC1WGD4
Whatmann blood collection	\$45.30	0.54	500	\$0.0904	184.8	\$8,252.98 https://www.sigmaaldrich.com/US/en/products/details/beta-1001018
Chemicals (PBS, Tween 20, BSA, water)	\$1,000.00	50.00	46200	\$0.0216	2	\$2,000.00 Variety of cheap chemicals
BSA-biotin	\$129.00	5.00	4620	\$0.0279	20	\$2,580.00 https://www.sigmaaldrich.com/US/en/products/details/bsa
Consumables (appendix, falcon tubes, tips)	\$1,000.00	50.00	46200	\$0.0216	2	\$2,000.00 Variety of cheap consumables
Tempets (Clips)	\$12,000.00	100.00	92400	\$0.1299	1	\$12,000.00 https://www.formalabs.com/product-category/tempets/
Tempets (Other consumables)	\$2,000.00	100.00	92400	\$0.0216	1	\$2,000.00 https://www.formalabs.com/product-category/tempets/
Isotope conjugation kit (for patient barcode)	\$1,200.00	2.00	1848	\$0.6494	50	\$60,000.00 https://www.illumina.com/Store/Storefront/Consumables/Reagents/Cytometry/MassCytometry/Labeling/Kit/Toolkit/US
Isotope conjugation kit (for sample barcode)	\$600.00	2.50	1848	\$0.2447	50	\$30,000.00 https://www.illumina.com/Store/Storefront/Consumables/Reagents/Cytometry/MassCytometry/Labeling/Kit/Toolkit/US
Protein (each)	\$500.00	50.00	46200	\$0.0108	40	\$20,000.00 Assuming \$500 per 100 ug of protein, prices might vary
Total cost						\$190,854.98
Cost per assay						\$2.07
Cost per sample						\$0.10
Cost per test in sample						

Figure R6. Cost of materials for each mass cytometry run of 924 samples and 20 targets.

Comparing the mass cytometry assay to other available technologies that can be used in diagnostic labs is challenging given stark differences in implementation and performance. Nevertheless, we can draw certain conclusions by comparing known costs of commercially available products to the costs described in the previous paragraph:

- ELISA is the most used strategy in clinical labs for detecting and quantifying antibodies. A commercial single-plex ELISA plate for a research lab has a typical cost of \$500 and it enables running 96 samples (including controls), with the cost of materials being roughly **\$5.21 per analyte per sample**. Purchasing and maintenance costs of a single microplate reader for analysis of ELISA plates is much lower than the cost of a mass cytometer, however, implementing large scale population studies with ELISA at the scale it would be possible with mass cytometry would require a significant investment in multiple microplate readers. Moreover, implementing large scale population studies with ELISA at the scale it would be possible with mass cytometry would also require the annual full-time salaries of a large number of technicians.
- Luminex assays are also used in clinical labs for multiplex detection of analytes. A commercial 11-plex ProcartaPlex Human Coronavirus Ig Total Panel (ThermoFisher, #EPX110-16000-901) plate for a research lab has a typical cost of \$2,365 and it enables running 96 samples (including controls), with the cost of materials being roughly **\$2.24 per analyte per sample**. Implementing large scale population studies with Luminex at the scale it would be possible with mass cytometry would also require the annual full-time salaries of several technicians, although the number would be lower for Luminex compared to ELISA. Capital and maintenance costs of the xMAP INTELLIFLEX Luminex system would also be significant.

Overall, serology by mass spectrometry analysis of beads has the potential to be a cost-effective technique in large-scale studies. It also strengthens the strategies for pandemic preparedness and response by diversifying the required raw materials and readouts.

In the revised version of the manuscript, we have included **Supplementary Note 1** (page 14, line 367), which includes the cost of materials for 100 mass cytometry runs of 924 samples and 20 targets, and we have included the following sentence in the Discussion section of the manuscript (page 14, line 366):

“The ability to run tens of thousands of assays in parallel makes it a cost-effective technology (**Supplementary Note 1**). Moreover, compared to other strategies used in diagnostic labs such as ELISA and Luminex assays, the hands-on time of operators running the same number of assays is substantially reduced.”

Missing or unclear details about statistics, protocols or materials

1. In the "Methods" section, no details about flow cytometry and mass cytometry methods are provided.

We thank the Reviewer for pointing this out. Details about flow cytometry and mass cytometry were given in the Methods section in the following subsections: “Two-bead assay by flow cytometry”, “Two-bead assay by mass cytometry”, “Wash free assay by flow cytometry”, and “Wash free assay by mass cytometry”. However, certain information regarding the acquisition was missing. In the revised version of the manuscript, we have included the following edits:

- In the subsection “Two-bead assay by flow cytometry”, we have included the following paragraph (page 19, line 547):

“For each sample, acquisition was based on a specific volume, either 20 μ L or 50 μ L. Fluorescent Blue and Nile Red beads were initially identified and gated using forward and side scatter channels and subsequently gated based on their fluorescence signal. This volume-based acquisition approach enabled the counting of the total number of beads in each sample volume. The counts obtained were then analyzed and plotted using R.”

- In the subsections “Two-bead assay by mass cytometry” and “Wash free assay by mass cytometry”, we have included the following paragraph (page 20, line 577 and page 21, line 616):

“The beads were analyzed using a CyTOF 2 mass cytometer (Fluidigm), at a rate of 500 events per second. The focus of this analysis was on isotopes within the mass range of 159 to 176.”

- In the subsection “Wash free assay by flow cytometry”, we have included the following paragraph (page 20, line 595):

“For each sample, a volume of 50 μ L was analyzed. Beads were first distinguished and gated using forward and side scatter channels, followed by analysis based on the fluorescence intensity of anti-IgG conjugated with Alexa Fluor 647. Subsequent data analysis was performed utilizing R software.”

2. The data set in Mendeley Data linked in the "Data availability" section is not public, and no access details have been provided as far as I am aware.

We apologize for this issue. Raw data, code, and a debarcoding demo related to this manuscript it is now public at <https://doi.org/10.5281/zenodo.10822264>.

Stylistic issues

1. The sentence in lines 108-111 needs to be revised.

We thank the Reviewer for pointing this out. In the revised version of the manuscript, the sentence:

“First, we performed mass cytometry analysis of streptavidin-coated beads loaded with biotinylated protein carrier conjugated to Dysprosium 162 (^{162}Dy)-containing maleimide-polymer showed an efficient detection of the stable lanthanide isotope and uniform labeling (**Fig. 1b**).”

Has now been re-written to (page 3, line 109):

“We conjugated the stable isotope Dysprosium 162 (^{162}Dy) to biotinylated bovine serum albumin (BSA) and loaded it to streptavidin-coated beads. Mass cytometry of these beads reveal a high load of isotopes per bead and a uniform labeling (**Fig. 1b**).”

2. As is frequently the case, the manuscript comes with nicely designed, but somewhat overloaded figures. The text for some panels is impossible to decipher even when enlarging (e.g., Figure 3f, especially the legend for the color scheme). Therefore, the font size must be increased to improve legibility, or the content needs to be split into several figures.

We also thank the Reviewer for pointing this out. We have increased the font size across most of the figures and supplemental figures to improve legibility.

Drainas, et al.

References

Han, *et al.* Metal-Isotope-Tagged Monoclonal Antibodies for High-Dimensional Mass Cytometry. *Nature Protocols*. **13**(10): 2121–48 (2018).

Tricot, S. *et al.* Evaluating the efficiency of isotope transmission for improved panel design and a comparison of the detection sensitivities of mass cytometer instruments. *Cytometry A*. **87**, 357–368 (2015).